# Neuron-specific RNA-sequencing reveals different responses in peripheral neurons after nerve injury

Sara Bolívar[1,2], Elisenda Sanz[1], David Ovelleiro[3], Douglas W Zochodne[4], Esther Udina[1,2]*

[1]Institute of Neurosciences, and Department Cell Biology, Physiology and Immunology, Universitat Autònoma de Barcelona, Bellaterra, Spain; [2]Centro de Investigación Biomédica en Red sobre Enfermedades Neurodegenerativas, Instituto de Salud Carlos III, Madrid, Spain; [3]Peripheral Nervous System, Vall d'Hebron Institut de Recerca (VHIR), Vall d'Hebron Hospital Universitari, Vall d'Hebron Barcelona Hospital Campus, Barcelona, Spain; [4]Division of Neurology, Department of Medicine and the Neuroscience and Mental Health Institute, University of Alberta, Edmonton, Canada

**Abstract** Peripheral neurons are heterogeneous and functionally diverse, but all share the capability to switch to a pro-regenerative state after nerve injury. Despite the assumption that the injury response is similar among neuronal subtypes, functional recovery may differ. Understanding the distinct intrinsic regenerative properties between neurons may help to improve the quality of regeneration, prioritizing the growth of axon subpopulations to their targets. Here, we present a comparative analysis of regeneration across four key peripheral neuron populations: motoneurons, proprioceptors, cutaneous mechanoreceptors, and nociceptors. Using Cre/Ai9 mice that allow fluorescent labeling of neuronal subtypes, we found that nociceptors showed the greater regeneration after a sciatic crush, followed by motoneurons, mechanoreceptors, and, finally, proprioceptors. By breeding these Cre mice with Ribotag mice, we isolated specific translatomes and defined the regenerative response of these neuronal subtypes after axotomy. Only 20% of the regulated genes were common, revealing a diverse response to injury among neurons, which was also supported by the differential influence of neurotrophins among neuron subtypes. Among differentially regulated genes, we proposed MED12 as a specific regulator of the regeneration of proprioceptors. Altogether, we demonstrate that the intrinsic regenerative capacity differs between peripheral neuron subtypes, opening the door to selectively modulate these responses.

*For correspondence:
esther.udina@uab.cat

Competing interest: The authors declare that no competing interests exist.

## eLife assessment

The **valuable** findings in this study show that subpopulations of peripheral sensory neurons display different capacities for regeneration after a similar injury. Nociceptor neurons have greater regeneration over mechanoreceptor, proprioceptors and motor neurons. This differential responsiveness of neuronal subtypes was traced to activation of different transcriptional programs, which were carefully analyzed and quantitated, resulting in **solid** evidence for the conclusions.

## Introduction

Peripheral nerve injuries result in the loss of motor, sensory, and autonomic function of denervated targets. Although peripheral neurons can regenerate after injury, most nerve lesions result in an

incomplete functional recovery. Strategies to promote regeneration often do not take into account the heterogeneity of peripheral neurons. Motor and sensory neurons are very different in terms of function, molecular identity, and their response to injury. Moreover, peripheral sensory neurons have a wide range of functions and target organs, and single-cell RNA-sequencing analyses have distinguished up to 11 subtypes in dorsal root ganglia (DRGs) (*Usoskin et al., 2015*; *Renthal et al., 2020*). Whereas the majority of DRG sensory neurons are cutaneous afferents, some innervate muscle (mainly proprioceptors) or other organs. Since neuron subtypes differentially respond to environmental cues and might activate distinct intrinsic regenerative programs, there is a need to investigate the regenerative response of specific neuron subtypes after a nerve injury.

Attempts to study variations in axon regeneration among subtypes of peripheral neurons have yielded some contradictory results. The finding that sensory neurons regenerate faster than motoneurons has robust evidence (*Dolenc and Janko, 1976*; *Madorsky et al., 1998*; *Suzuki et al., 1998*; *Kawasaki et al., 2000*; *Negredo et al., 2004*; *Brushart et al., 2020*). In addition, unmyelinated fibers have been described to recover their function earlier than myelinated fibers (*Navarro et al., 1994*). In contrast, alternative work suggests that myelinated fibers regenerate at the same speed as unmyelinated fibers (*Lozeron et al., 2004*; *Moldovan et al., 2006*) or even that motoneurons regenerate better than sensory neurons (*da Silva et al., 1985*). Given that sensory neurons comprise both myelinated and unmyelinated fibers, it is important to clarify the distinctions involving regeneration of different neuron subtypes.

After axotomy, peripheral neurons activate several signaling mechanisms such as the Ras/Raf/MAPK pathway (*Sheu et al., 2000*; *Obata et al., 2003*; *Agthong et al., 2006*), the phosphoinositide 3-kinase/protein kinase B (Akt) pathway (*Kimpinski and Mearow, 2001*; *Murashov et al., 2001*; *Edström and Ekström, 2003*), the c-Jun N-terminal kinase (JNK)/c-Jun pathway (*Kenney and Kocsis, 1998*), or the cAMP/protein kinase A/cAMP responsive element binding protein pathway (*Gao et al., 2004*; *Chierzi et al., 2005*). All these signaling pathways activate regeneration-associated genes, including *Gap43* (*Van der Zee et al., 1989*; *Kawasaki et al., 2018*; *Mason et al., 2022*). This process enables neurons to switch to a pro-regenerative state in which axon regrowth is permitted. However, as neuron subtypes are thought to regenerate at different rates, the transcriptional regeneration mechanism activated by these neurons might be expected to differ. In fact, some regeneration-associated pathways have been shown to have specific relevance in neuron subtypes. For instance, medium-to-large DRG neurons show an increased activation of extracellular signal-regulated protein kinase compared to small DRG neurons (*Obata et al., 2003*; *Obata et al., 2004*). Importantly, the RhoA/Rho-kinase pathway may have a different role in motor and sensory regeneration given evidence that its inhibition enhances motor but not sensory axon regeneration (*Joshi et al., 2015*). Understanding the intrinsic regenerative differences between neuron subtypes may allow fine-tuning of the specific regeneration of neuron subtypes, prioritizing the growth or guidance of specific subpopulations toward their target organs. Overall, neuron regeneration involves several levels of complexity driven by a large ensemble of molecules that may be challenging to differentiate among subtypes.

To address these challenges, we used two lines of genetically engineered mice that allowed interrogation of regenerative properties among peripheral neuron subtypes. First, we described the regeneration of specific neurons after injury using TdTomato Cre reporter mice (Ai9). Breeding these to mice expressing Cre recombinase under the control of specific promoters allowed us to study axonal regeneration in motoneurons (choline acetyltransferase, *Chat*), proprioceptors (parvalbumin, *Pvalb*), cutaneous mechanoreceptors (neuropeptide Y receptor Y2, *Npy2r*), and nociceptors (transient receptor potential vanilloid 1, *Trpv1*). We found significant differences in axonal regeneration of these neurons that suggested distinct intrinsic regenerative mechanisms. Second, we explored the neuron-specific gene expression of these neurons after a sciatic nerve injury in vivo. Cre-driver animals were crossed to Ribotag mice (*Sanz et al., 2009*), which express a modified ribosomal protein L22 (Rpl22) in a Cre-dependent manner. Through immunoprecipitation assays, ribosomes and the associated mRNA from specific cell populations can be isolated and sequenced. Using this approach, we were able to describe the gene expression patterns of motoneurons, proprioceptors, mechanoreceptors, and nociceptors after a nerve injury. Amidst this approach and data, we explored the role of one of the differentially expressed transcripts, mediator complex subunit 12 (*Med12*), a unique and unexplored protein, as a specific regulator of the regeneration of proprioceptors.

## Results

### Axon regeneration in motor and sensory neurons has different rates and patterns

A comparative study of regeneration in peripheral neuron subtypes is complex and requires specific markers to label neuron populations. We took advantage of genetic labeling to comparatively study four key peripheral populations. In our previous report, we characterized and validated ChAT-Cre/Ai9, PV-Cre/Ai9, and Npy2r-Cre/Ai9 mice for the study of motoneurons, proprioceptors, and cutaneous mechanoreceptors, respectively (*Bolívar and Udina, 2022*). Here, we added TRPV1-Cre/Ai9 mice for the study of nociceptors, as described previously (*Patil et al., 2018*). We corroborated that the labeled neurons in the DRG were mostly peptidergic (42.8 ± 1.2% co-labeling with CGRP) and non-peptidergic (30.1 ± 1.4% IB4-reactive) small-diameter neurons. Additionally, fluorescent axons were found in the skin as free endings (*Figure 1—figure supplement 1*).

The regeneration in these populations was established by crushing the sciatic nerve and counting the number of axons that had regenerated 7 and 9 days after the injury (*Figure 1A*). In control nerves, we found that nociceptors and mechanoreceptors were the populations with the most abundant axons, followed by proprioceptors and motoneurons. As the number of axons differ between populations in control conditions, we evaluated both the total number of regenerated axons (*Figure 1C*) and the relative regeneration compared to each control (*Figure 1D*). Seven days after the injury, the number of regenerated axons of the different subpopulation did not reach control values when assessed at a distance of 17 mm from injury site (*Figure 1C*). However, motoneurons and nociceptors achieved axon regrowth similar to controls at 12 mm from the injury site (*Figure 1C*, p>0.05 vs their control). In contrast, by 9 days after crush, we found that all axon populations had regenerated to numbers comparable to uninjured controls at 12 mm, and only motoneuron and nociceptor axons reached control values at 17 mm (*Figure 1C*, p>0.05 vs their control). Therefore, motoneurons and nociceptors were the populations that recovered their number of axons earlier. Despite this similarity, the pattern of regeneration differed between these populations. Motoneurons regrew significantly more axons at 12 mm 9 days after the injury than in control conditions (p=0.038), probably due to the collateral branching at the regenerative front, whereas nociceptors displayed an increase in axon number reaching a plateau at control levels. Since none of the populations reached control values at 17 mm and 7 days postinjury (dpi), we considered it the most discriminating evaluation point. At this time and distance, the relative number of regenerated motor axons was significantly lower than that of nociceptors (p=0.012), indicating that nociceptors were the population with a greater axonal regeneration (*Figure 1D*). Although non-significant, mechanoreceptors showed a tendency to regenerate proportionately more than proprioceptors, at both times and distances (*Figure 1D*). This pattern was maintained in DRG explants in vitro, where nociceptors extended significantly longer neurites than other sensory neurons and proprioceptors were the population with shorter neurites (*Figure 1—figure supplement 2*). Altogether, these results indicated that nociceptors had the greater regeneration, followed by motoneurons and, finally, cutaneous mechanoreceptors and proprioceptors.

### RNA isolation in Cre/Ribotag mice is neuron population-specific

We isolated the pool of actively translated mRNAs in specific neuron populations using the Ribotag assay (*Figure 2A*). We used the Cre-driver animals ChAT-Cre, PV-Cre, Npy2r-Cre, and TRPV1-Cre bred to Ribotag mice to specifically target the ribosomes of motoneurons, proprioceptors, cutaneous mechanoreceptors, and nociceptors, respectively. These animals showed the expression of the hemagglutinin (HA) ribosomal tag in neurons either in the DRG (co-labeling with β-tubulin) or the spinal cord (co-labeling with ChAT), with a similar pattern to their Cre/Ai9 counterparts (*Figure 2B and C*). After RNA isolation, RT-qPCR from immunoprecipitates (IPs) showed a several-fold enrichment of the cell type-specific transcripts *Chat*, *Pvalb*, *Npy2r*, and *Trpv1* in their respective IPs (*Figure 2D*). In contrast, the glial transcript Fabp7 (fatty acid binding protein 7) was depleted in all IPs, indicating that the ribosome isolation was neuron-specific.

The RNA-sequencing analysis showed that more than 3000 genes were differentially expressed in each subpopulation of neurons. Before further analysis of these data, we assessed its validity by checking the transcripts per million of several markers. We confirmed that IPs from motoneurons had enriched transcripts of the well-established cell type-specific markers, including *Chat*, *Mnx1*, *Isl1*, *Nefh*, and *Tns1* (*Figure 2—figure supplement 1*). *Tubb3* and *Uchl1*, which are pan-neuronal markers,

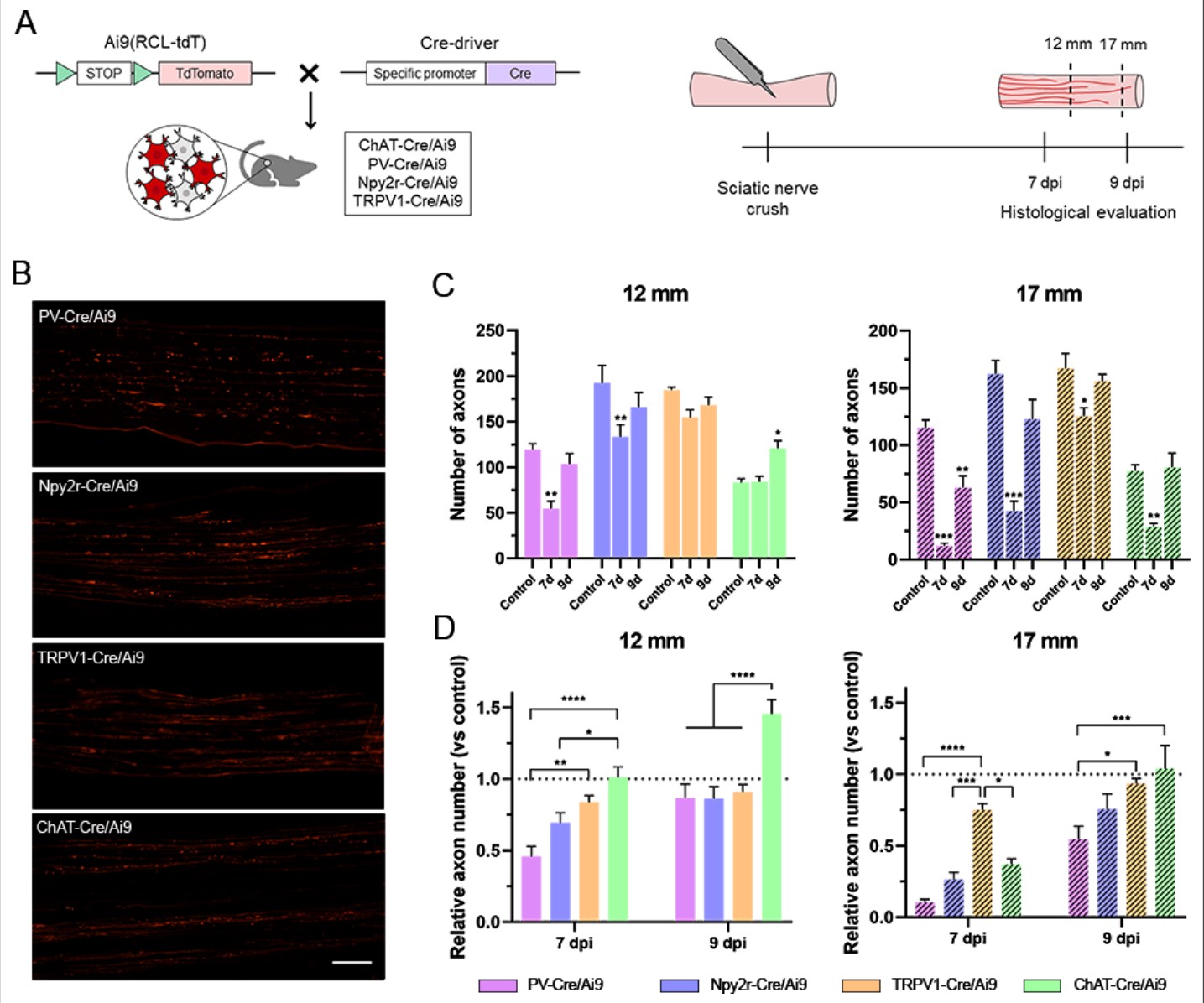

**Figure 1.** Regeneration rate of peripheral neurons. (**A**) Schematic representation of the transgenic mice and experimental design used in this study. (**B**) Microscope images of longitudinal sections of regenerating nerves in the different Cre/Ai9 animals 7 days after injury. Images show a representative example of distal regeneration at 9 days postinjury (dpi). (**C**) Quantification of the number of axons that regenerated at 12 mm (smooth bars) and 17 mm (stripped bars) at 7 or 9 dpi. Each color represents a different neuron subtype (purple: proprioceptors; blue: cutaneous mechanoreceptors; orange: nociceptors; green: motoneurons). *p<0.05, **p<0.01, ***p<0.001 vs each control group as calculated by two-way ANOVA followed by Bonferroni's correction for multiple comparisons. (**D**) Relative number of regenerated axons normalized by the control number of axons in each neuron type. *p<0.05, **p<0.01, ***p<0.001, ****p<0.0001 as calculated by two-way ANOVA followed by Bonferroni's correction for multiple comparisons. $n_{7\ days}$ = 7/group; $n_{9\ days}$ = 7/group; $n_{control}$ = 3 (each sensory group) or 6 (motoneurons). Scale bar: 100 µm.

The online version of this article includes the following figure supplement(s) for figure 1:

**Figure supplement 1.** Characterization of TRPV1-Cre/Ai9 mice.

**Figure supplement 2.** In vitro neurite extension in dorsal root ganglia (DRG) explants.

were enriched in all sensory neurons, whereas *Pvalb*, *Npy2r*, and *Trpv1* transcripts were enriched in proprioceptors, mechanoreceptors, and nociceptors IPs, respectively. As expected, *Nefh* was enriched in proprioceptors but depleted in nociceptors. The transcripts encoded by the glial genes *Gfap*, *Fabp7*, *Tmem119*, *Mobp*, *Olig1*, and *Vim* were depleted in IPs (***Figure 2—figure supplement 2***). Among IPs, we saw an enrichment of the transcripts for the regeneration markers *Gap43*, *Atf3*,

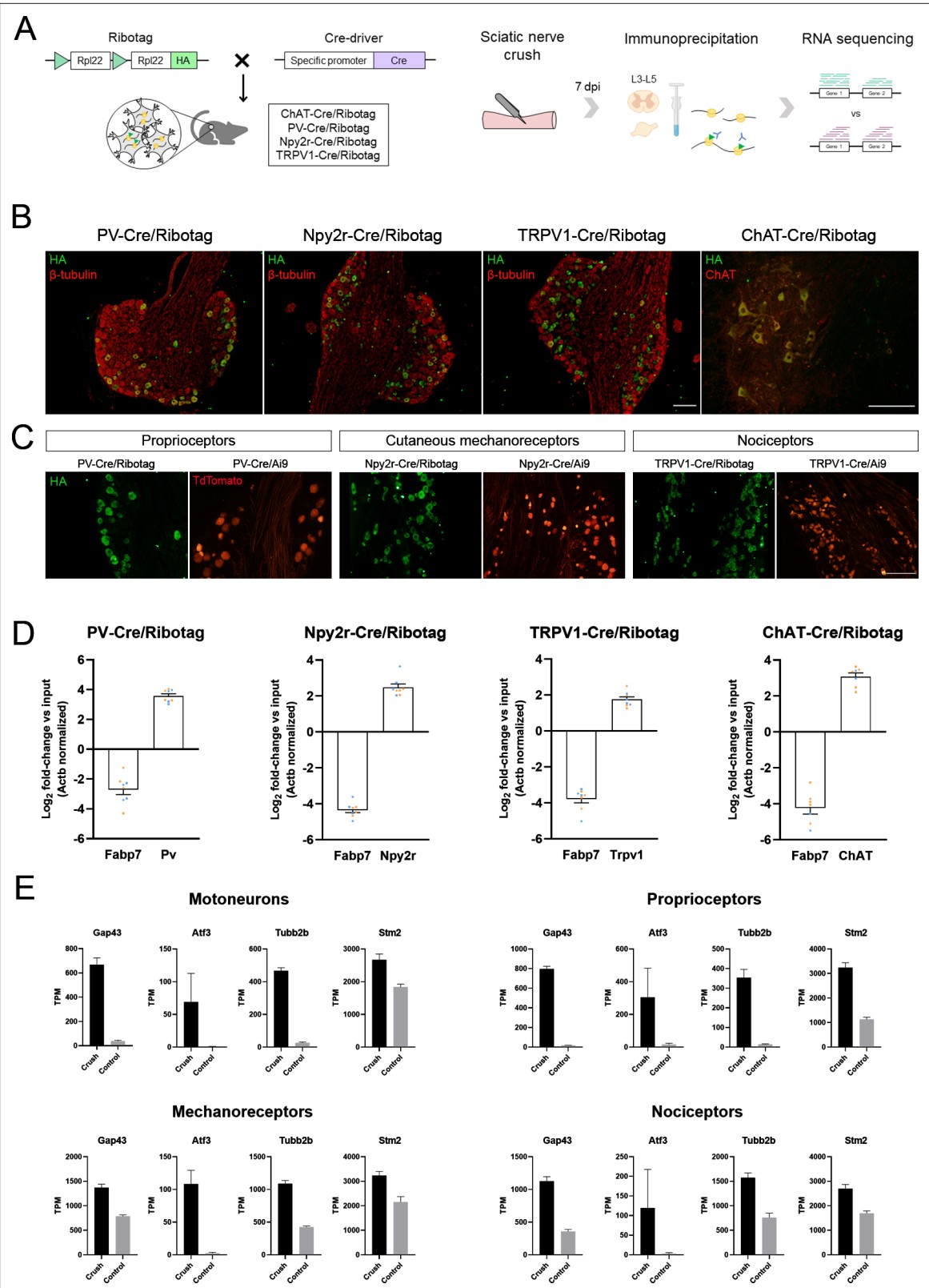

**Figure 2.** Validation of the specificity of PV-Cre/Ribotag, Npy2r-Cre/Ribotag, TRPV1-Cre/Ribotag, and ChAT-Cre/Ribotag mice. (**A**) Schematic representation of the mice and experimental design used in this experiment. (**B**) Immunostaining against hemagglutinin (HA, green) shows the expression of tagged ribosomes in neuronal cells in the dorsal root ganglia (DRG) (PV-Cre/Ribotag, Npy2r-Cre/Ribotag, and TRPV1-Cre/Ribotag) and in the motoneurons in the spinal cord (ChAT-Cre/Ribotag). In red, β-tubulin labels all cells in the DRG, and ChAT labels motoneurons in the spinal

*Figure 2 continued on next page*

*Figure 2 continued*

cord. (**C**) Cre/Ribotag mice expresses HA (in green) in a similar pattern to the expression of TdTomato in Cre/Ai9 mice (in red). (**D**) RT-qPCR reveals the enrichment of cell type-specific transcripts in each immunoprecipitate (*Pv, Npy2r, Trpv1, ChAT*) and the depletion of the glial transcript Fabp7 in immunoprecipitates from all neuron populations. In orange, samples from female mice; in blue, samples from male mice. (**E**) Transcripts per million (TPM) of regeneration markers. The expression of the transcripts *Gap43, Atf3, Tubb2b*, and *Stm2* is enriched in the immunoprecipitates from injured mice compared to control mice in all populations. Scale bar: 150 µm.

The online version of this article includes the following figure supplement(s) for figure 2:

**Figure supplement 1.** Transcripts per million (TPM) of cell type-specific transcripts in the spinal cord.

**Figure supplement 2.** Transcripts per million (TPM) of cell type-specific transcripts in the dorsal root ganglia (DRG).

**Figure supplement 3.** Principal component analysis (PCA) shows distinct transcriptome segregation according to the experimental group.

*Tubb2b*, and *Stm2* in lesioned neurons compared to controls in all neuron types (*Figure 2E*). Finally, principal component analysis revealed that samples clustered according to their experimental group (*Figure 2—figure supplement 3*).

## Peripheral neurons differentially activate regenerative programs

Compared to intact mice, a total of 3546, 4053, 3461, and 3281 were significantly differentially expressed in datasets from motoneurons, proprioceptors, mechanoreceptors, and nociceptors, respectively, 7 days after injury (*Supplementary file 2*, *Figure 3—figure supplements 1 and 2*). Many of these differentially expressed genes (DEGs) were enriched or depleted in more than one neuron type, whereas some of them were up- or downregulated in a specific neuron type (*Figure 3A and B*). Since there was an elevated number of DEGs in each condition, we focused on the ones with the highest fold-change in each neuronal population. *Figure 3C–F* shows the DEGs that were significantly up- or downregulated in each population with a log$_2$ fold-change above 4 or below –4 but were regulated in the opposite direction in the other populations or the fold-change was lower than |1|. Some of these genes are particularly important for their potential implication in specific regeneration. For instance, we found that *Ngfr* (p75NTR), a neurotrophin receptor, was specifically enriched in lesioned motoneurons. This upregulation was confirmed by immunohistochemistry (*Figure 3—figure supplement 3*). On the other hand, proprioceptors were the only population significantly upregulating *Nrp1* and *Nrp2* after injury, which are cell surface receptors for class 3 semaphorins. In nociceptors and cutaneous mechanoreceptors, some of the upregulated gens, such as *Il6ra* and *Atf2*, respectively, may be related with hyperalgesia. Increased number of ATF2 positive nuclei were also observed by immunohistochemistry in cutaneous mechanoreceptor neurons (*Figure 3—figure supplement 4*).

In terms of specific regeneration, we subdivided peripheral neurons into 'muscle neurons' or 'cutaneous neurons'. This allowed us to study potential candidates that could prioritize the regeneration of groups of neurons toward the appropriate nerve branch required to reach their original target organs. We considered motoneurons and proprioceptors as muscle neurons, since these innervate the muscle, and cutaneous mechanoreceptors and nociceptors as cutaneous neurons because their most common target is the skin. In cutaneous neurons, we found 20 genes with a log$_2$ fold-change above |2| that were not changed in muscle neurons or were regulated in the opposite direction in both groups (*Figure 3G*). Interestingly, only 2 of these genes were downregulated in cutaneous neurons: *Med12* and *Cacna1b*. In muscle neurons, we found 15 DEGs with a fold-change above |2| that were not changed in cutaneous neurons (*Figure 3H*). Finally, we evaluated the most differentially regulated genes in the three types of sensory neurons compared to motoneurons (*Figure 3I*). All these genes can be potential candidates to modulate the specific regeneration of these groups of neurons.

## Peripheral neuron subtypes differentially activate regenerative pathways

A combination of ontologies and databases were used in the enrichment analysis and the data are available in Supplementary material (*Supplementary file 3*). The caveat in these analyses is that many genes can be involved in multiple biological pathways and these databases might not entirely reflect the complete functional diversity of proteins. However, they provide a first approach for the interpretation of complex biological data. Gene Ontology (GO) analysis revealed significantly enriched terms associated with regeneration in all neuronal populations following injury, including cell projection

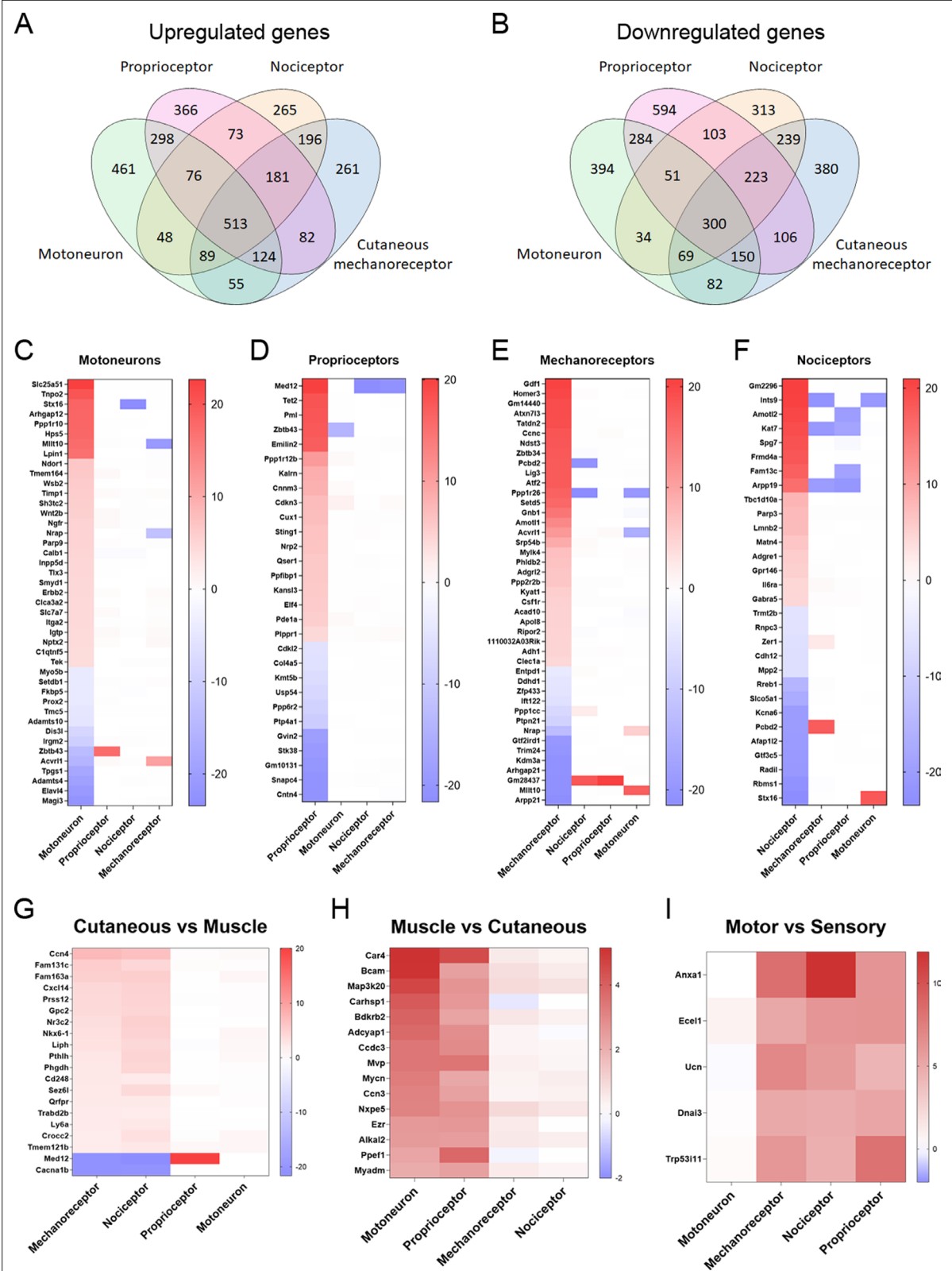

**Figure 3.** Differentially expressed genes (DEGs) vary between neuronal subtypes. (**A–B**) Number of genes that were commonly upregulated or downregulated between populations or uniquely expressed in one of the neuron subtypes. (**C–F**) Significantly up- or downregulated genes in each population with a $log_2$(fold-change) ($log_2$FC) above 4 or below −4. The genes shown in these graphs exclude those that are significantly regulated

*Figure 3 continued on next page*

*Figure 3 continued*

above |1| in the same direction in two or more neuron types. (**G–I**) DEGs in groups of neuronal populations. DEGs with a log$_2$(FC) above 2 or below −2 in cutaneous neurons (**G**), muscle neurons (**H**), or sensory neurons (**I**) are plotted from more upregulated to more downregulated.

The online version of this article includes the following figure supplement(s) for figure 3:

**Figure supplement 1.** Volcano plots showing genes significantly up- or downregulated in each population.

**Figure supplement 2.** Hierarchical clustering of the 80 most variable genes in each population.

**Figure supplement 3.** Overexpression of p75 on motoneurons 7 days after crush injury.

**Figure supplement 4.** Overexpression of AFT2 on cutaneous mechanoreceptors 7 days after crush injury.

organization, axon guidance, and Ras protein signal transduction (*Figure 4A*). Kyoto Encyclopedia of Genes and Genomes (KEGG) analysis showed that all neurons activate common pathways such as focal adhesion, MAPK, and cAMP signaling pathways (*Figure 4B*). There were some interesting GO terms and KEGG pathways that were differentially regulated between neuron types. Muscle neurons (motoneurons and proprioceptors) significantly regulated the ErbB and VEGF signaling pathways, which are potentially relevant for regeneration (*Figure 4C*). Importantly, GO analysis revealed a specific enrichment of the semaphorin-plexin pathway in these two neuron subtypes, a crucial pathway in neuron guidance during development and regeneration (*Figure 4C*). Cutaneous neurons showed a specific enrichment in cyclic guanosine monophosphate-dependent protein kinase (PKG) and aldosterone signaling pathways (*Figure 4C*). Other pathways were significantly regulated in specific neuron populations, such as the peroxisome proliferator-activated receptor or the AMP-activated protein kinase signaling in motoneurons. These two pathways are involved in many cell processes including metabolism, cell survival, and neuroprotective functions and were not activated in sensory neurons. JNK pathway, which has been associated to neuropathic pain, was found enriched only in nociceptors (*Figure 4D*). This data highlights the heterogeneity in the regenerative response of the distinct peripheral neurons after a nerve injury.

## Neurotrophic factors differentially influence sensory neurite outgrowth

Neurotrophins bind to Trk receptors, which are known to be differentially expressed in sensory neuron subpopulations. In our RNA-sequencing analysis, we found a differential expression of these receptors in our populations (*Supplementary file 1*). Therefore, we aimed to corroborate the differential effect of NGF, BDNF, and NT-3 (which bind TrkA, TrkB, and TrkC, respectively) in our three sensory neuron populations. First, we examined neurite extension in the DRG by labeling neurites with a pan-neuronal marker, PGP9.5 (*Figure 5—figure supplement 1*). We found that 10 ng/mL NGF produced the larger increase in neurite length, being the mean longest neurite of 927.5±85.0 µm (p<0.0001 vs control 441.7±32.0 µm). We also observed a significant increase in neurite length with 10 ng/mL BDNF (771.1±54.8 µm, p<0.0001) and 10 ng/mL NT-3 (64.0±48.6 µm, p=0.044).

Next, we evaluated if these neurotrophic factors had a differential effect in distinct neuronal subtypes. The neurite length of proprioceptors was only significantly increased when adding NT-3 (282.8±57.5 µm vs control 121.8±10.1 µm, p=0.011), but not NGF or BDNF (*Figure 5B*). For cutaneous mechanoreceptors, we did not find a significant increase in neurite outgrowth with any of the neurotrophic factors used (*Figure 5C*). However, NGF showed a tendency to increase the longest neurite in this population (p=0.052). Contrarily, both NGF and BDNF significantly increased the neurite length of nociceptors (973.5±63.8 µm, p<0.0001 for NGF; 780.1±54.4 µm, p=0.002 for BDNF, both vs control 479.2±45.2 µm) (*Figure 5D*). The values of neurite length were normalized to their control groups for each neuron population and expressed as a fold-change to allow the direct comparison between groups (*Figure 5E*). Interestingly, we found that cutaneous mechanoreceptors and nociceptors had a similar pattern, which was clearly different from proprioceptors. The most specific growth factor for cutaneous neurons was NGF, which increased neurite length 1.69-fold in mechanoreceptors and 2.03-fold in nociceptors. In both cases, this increase was significantly higher than in proprioceptors, which showed a 0.45-fold change (p=0.007 vs mechanoreceptors; p<0.0001 vs nociceptors). In contrast, the most specific factor for muscle sensory neurons was NT-3. Proprioceptors showed a 2.32-fold increase in neurite length in opposition to mechanoreceptors (0.8-fold change, p=0.002) and nociceptors (1.13-fold change, p<0.0001). Taken together, the findings supported established links between

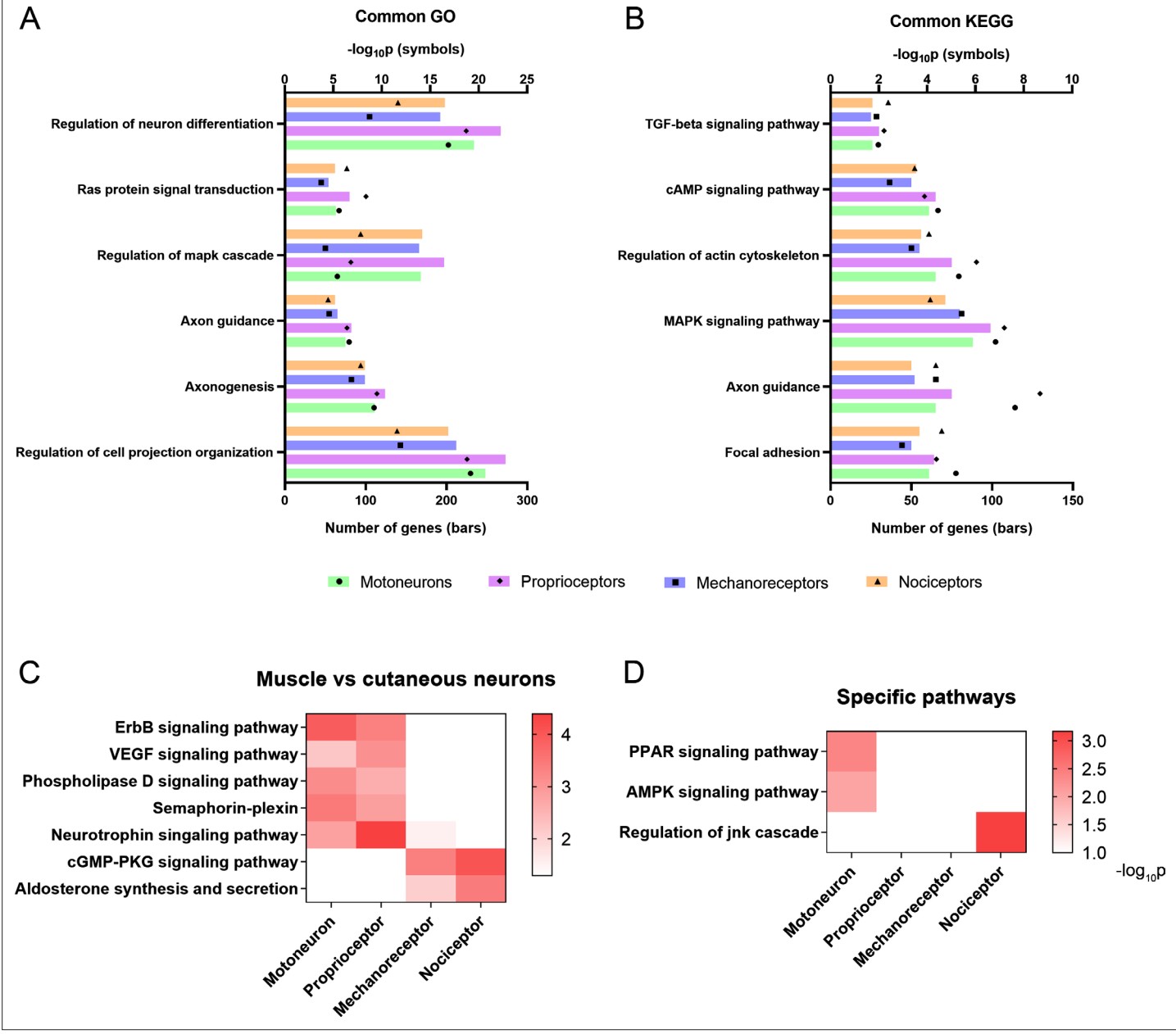

**Figure 4.** Activation of relevant pathways in axon regeneration. (**A**) Six of the most relevant Gene Ontology (GO) processes that are significantly activated by all the studied neurons after injury. (**B**) Selection of relevant pathways enriched in neurons after injury according to the Kyoto Encyclopedia of Genes and Genomes (KEGG) database. (**C**) Some GO processes (semaphoring-plexin) and KEGG pathways (all the others) enriched in specific neuron groups. (**D**) Examples of relevant pathways enriched in a specific neuron subtype. TGF: transforming growth factor, cAMP: cyclic adenosine monophosphate, MAPK: mitogen-activated protein kinases, VEGF: vascular endothelial growth factor, cGMP-PKC: cyclic guanosine monophosphate-protein kinase C, PPAR: peroxisome proliferator-activated receptor, AMPK: AMP-activated protein kinase, JNK: c-Jun N-terminal kinase.

specific neurotrophin molecules and sensory neuron subtypes (***Ernsberger, 2009***) and validated the specificity of our isolation approach.

## MED12 specifically inhibits neurite extension of proprioceptors

Results of the RNA-sequencing showed that expression of *Med12* was upregulated in proprioceptors (20.1 log$_2$ fold-change) and downregulated in mechanoreceptors and nociceptors (−21.0 and −21.6 log$_2$ fold change, respectively) after injury. In cancer cell lines, *Med12* has been described to inhibit TGF-β signaling (***Huang et al., 2012***), an important pathway implicated in regeneration. Thus,

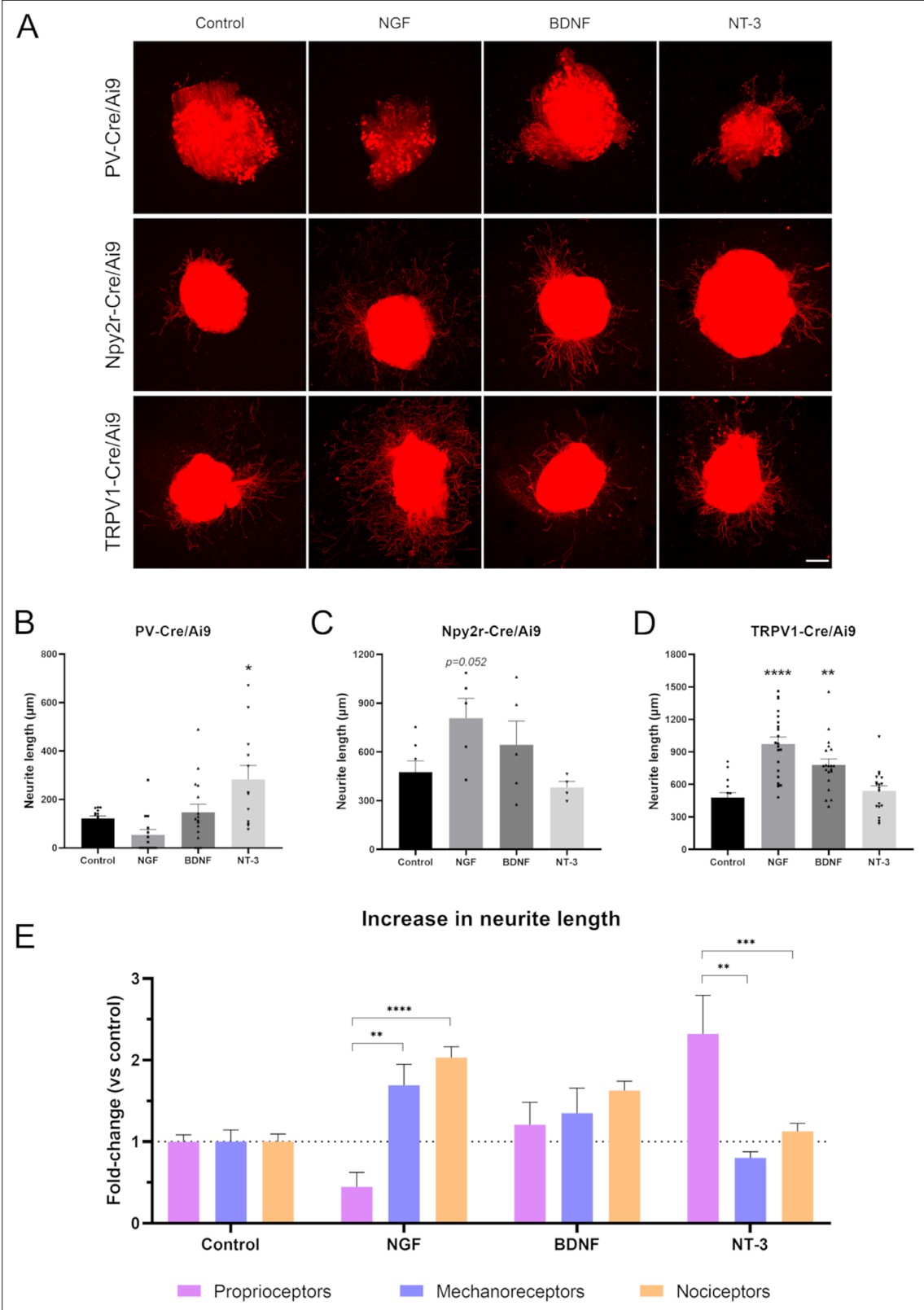

**Figure 5.** Neurite extension in PV-Cre/Ai9, Npy2r-Cre/Ai9, and TRPV1-Cre/Ai9 dorsal root ganglia (DRG) explants. (**A**) Microscope images of the neurite outgrowth of each neuron population in explants with nerve growth factor (NGF), brain-derived neurotrophic factor (BDNF), and neurotrophin-3 (NT-3). (**B–D**) Quantification of the longest neurite in each condition. *p<0.05 vs control as calculated by one-way ANOVA followed by Bonferroni's multiple comparisons test (PV-Cre/Ai9 and Npy2r-Cre/Ai9) or by Kruskal-Wallis test followed by Dunn's multiple comparisons test (TRPV1-Cre/Ai9).

*Figure 5 continued on next page*

*Figure 5 continued*

(**D**) Comparison of the increase in neurite length in proprioceptors, mechanoreceptors, and nociceptors. The increase is plotted as a fold-change vs each control to compare the effect of the neurotrophic factors. **p<0.01, ***p<0.001, ****p<0.0001 vs proprioceptors as calculated by a two-way ANOVA and followed by a Tukey's post hoc test. Scale bar: 200 μm.

The online version of this article includes the following figure supplement(s) for figure 5:

**Figure supplement 1.** Neurite extension in the dorsal root ganglia (DRG).

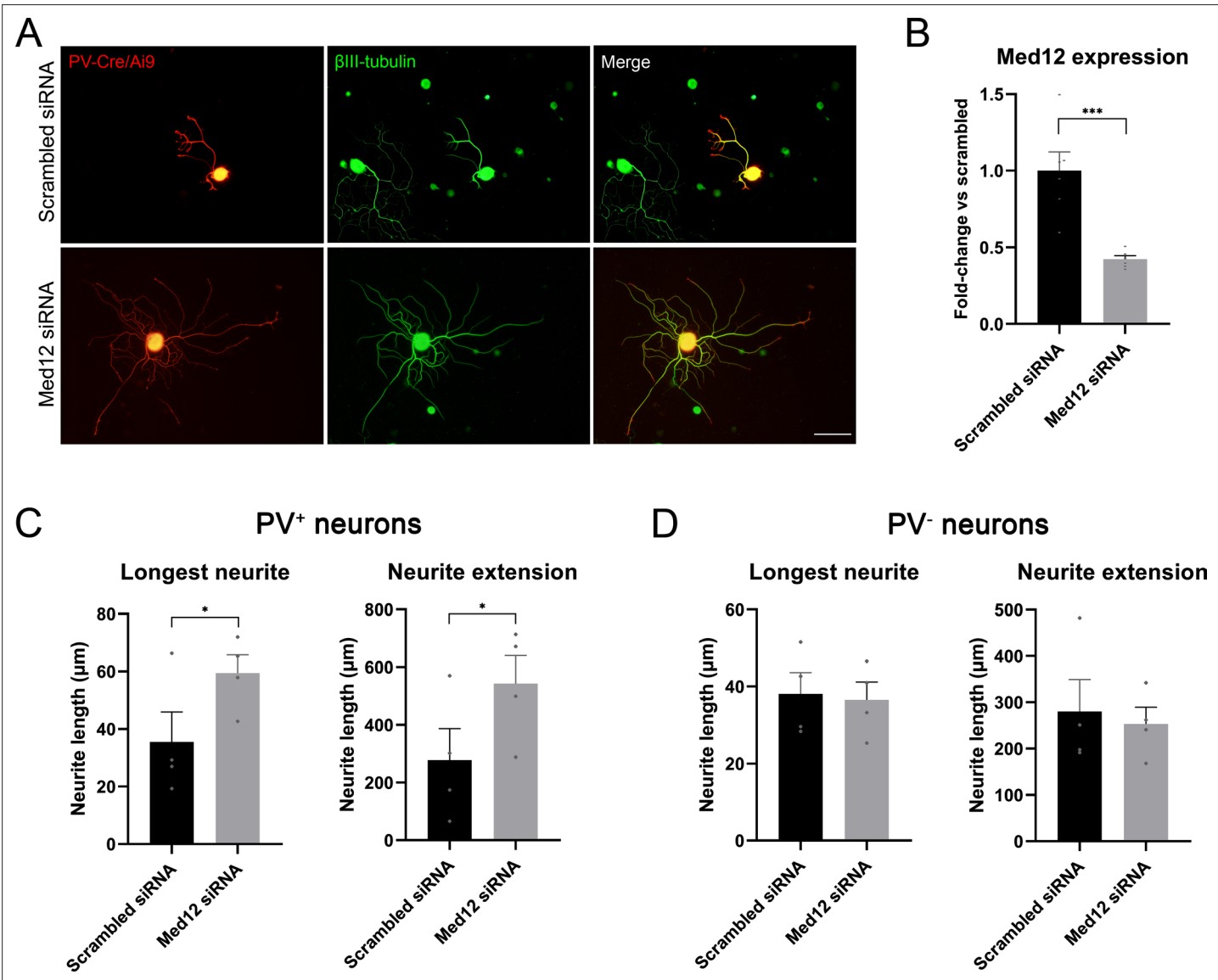

**Figure 6.** Knockdown of *Med12* in dissociated dorsal root ganglia (DRG) cultures. (**A**) Microscope images from proprioceptors (in red) in cultures with scrambled or Med12 siRNA. βIII-tubulin was used as a pan-neuronal marker (in green). (**B**) *Med12* expression measured by qPCR and expressed as fold-change vs scrambled. (**C**) Quantification of the neurite length in PV⁺ neurons (fluorescent neurons in PV-Cre/Ai9). The mean of the longest neurite and the total length per neuron in each culture are plotted. (**D**) Quantification of neurite length in PV⁻ neurons, labeled by βIII-tubulin. *p<0.05, ***p<0.001 as calculated by t-test. Scale bar: 100 μm.

The online version of this article includes the following figure supplement(s) for figure 6:

**Figure supplement 1.** Gene expression of TGF-β pathway mediators measured by qPCR and expressed as fold-change vs scrambled.

we thought to determine its role in the regeneration of specific neuronal subtypes in vitro in naïve neurons. A mix of four different siRNAs was used to target *Med12* and its expression was evaluated by qPCR. We found that Med12 was significantly downregulated using this approach (p=0.001), with a 57% decrease compared to the scrambled siRNA condition (*Figure 6B*).

Next, we used the fluorescence in PV-Cre/Ai9 mice to specifically evaluate neurite extension of proprioceptors (*Figure 6C and D*). We found that the longest neurite and the total neurite extension per neuron were significantly increased in the knockdown group (p=0.039 and p=0.014, respectively). To ensure that this effect was neuron type-specific, we analyzed neurite extension in non-proprioceptive neurons (PV⁻) labeled with βIII-tubulin. We did not find a significant change in the length of the longest neurite nor in total neuron extension, thus suggesting that MED12 specifically regulated proprioceptive neurite extension. We evaluated the expression of some genes implicated in the TGF-β pathway by qPCR (SMAD7, CRMP2, DAB2, SERPINE, TGFBR2), but we did not find a significant difference between the control and knockdown group (*Figure 6—figure supplement 1*).

## Discussion

Peripheral neurons are heterogeneous and have distinct functions and target organs. Here, we show that, after axotomy, there is a specific response to injury that strongly varies between neuron types. We found differences in the regeneration ability of neuron subtypes after nerve injury, which were associated with an activation of distinct gene expression patterns. Previous studies have analyzed the transcriptome of peripheral neurons or the whole DRG after injury (*Kaly et al., 2018*; *Lemaitre et al., 2020*), but our results highlight the importance of the study of regeneration in specific neuron subtypes to improve specific regeneration after nerve injury.

In our crush study, we found that nociceptors were the peripheral population with greater axon growth. Unmyelinated fibers have been described to recover their function earlier than myelinated fibers (*Navarro et al., 1994*). As our population of nociceptors includes mostly unmyelinated fibers, a faster regeneration rate of these fibers could explain the advantage of these neurons over other populations. Similarly, the regeneration rate of sensory neurons has been reported to be faster than that of motoneurons (*Dolenc and Janko, 1976*; *Madorsky et al., 1998*; *Suzuki et al., 1998*; *Kawasaki et al., 2000*; *Negredo et al., 2004*; *Brushart et al., 2020*). Most of these studies, however, use functional assessments to determine the speed of regeneration, which can be influenced by the reinnervation capacity of the different fibers. Motoneurons also showed heightened growth since their axons reached control values at the same time as nociceptors. This is in accordance with previous reports stating that myelinated sensory fibers can regenerate at the same speed as unmyelinated fibers (*Lozeron et al., 2004*). However, this population had a regeneration pattern distinct from the other neurons. At 12 mm, we found a significantly higher number of axons than in the control group which could be explained by the presence of regenerative collaterals. In fact, we have previously described that motoneurons extend more regenerative collaterals than proprioceptors, confirmed by the current work (*Bolívar and Udina, 2022*). Cutaneous mechanoreceptors reached values of axon regrowth comparable to controls later than nociceptors and motoneurons. This finding differs from a previous work identifying that this population regenerates better than motoneurons after a femoral transection (*Bolívar and Udina, 2022*). However, the approach and the territory used to assess regeneration differed, previously using retrotracers to evaluate numbers of regenerating femoral neurons, compared to direct counts of sciatic regenerating axons here. Since our cutaneous mechanoreceptors include myelinated and unmyelinated fibers, a more variable regeneration rate can be expected in these animals. We speculate that unmyelinated Npy2r⁺ fibers may regenerate faster than motoneurons, but this might be masked by the motor regenerative collaterals and the heterogeneity in Npy2r⁺ neurons. Finally, proprioceptors showed more limited regeneration. After the crush injury, this population exhibited a relatively lower proportion of axons compared to the other populations, both in terms of time and distance, which is in agreement with our previous experiments (*Bolívar and Udina, 2022*).

Altogether these results indicate that sensory regeneration is inversely proportional to the fiber size: large, myelinated neurons regenerate less robustly than smaller and less myelinated neurons. Previous authors described this phenomenon by using cuff electrodes after a crush in the tibial nerve of cats (*Krarup et al., 1989*) or with retrograde tracers after transection (*Negredo et al., 2004*). As for motoneurons, our results suggest that, despite their large fiber size, their regeneration exceeds large-size DRG neurons but not small-size sensory neurons.

Axotomy initiates a massive response in neurons which involves several signaling pathways and the upregulation of RAGs. In our RNA-sequencing analysis, we saw a common upregulation of 515 transcripts and a downregulation of 300 transcripts across all studied populations. Among these, we found *Gap43, Tubb2a, Sprr1a, Jun, Stat3, Sox11, Atf3, Hspb1, Gfra1,* and *Gal*. All these genes are associated with regeneration after injury and growth cone dynamics (*Tedeschi, 2011*). Surprisingly, most induced transcripts differed between neuronal populations: 75–80% of DEGs were specifically regulated in a single neuron type or in groups of neurons. These are potential factors explaining the differences in regenerative capacity between neurons subtypes. Based on these results, we studied two strategies to selectively modulate neurite growth in vitro: addition of neurotrophic factors and analysis of the impact of knockdown of a specific DEG, namely *Med12*.

Peripheral neurons express distinct Trk receptors according to their subtype. Broadly, peptidergic neurons express the NGF receptor TrkA, myelinated Aβ fibers have the BDNF receptor TrkB, and proprioceptors express NT-3 receptor TrkC. These neurotrophins have been described to be involved in the survival and/or neurite extension of peripheral neuron subpopulations (*DiStefano et al., 1992*; *Terenghi, 1999*). For this reason, we aimed to corroborate that different neurotrophic factors would elicit a distinct response in our sensory neuron subpopulations. When neurotrophic factors were analyzed overall, evaluating the growth of all neurites, we found that NGF, BDNF, and NT-3 promoted sensory outgrowth. This is in agreement with previous reports showing an increased axon elongation in explants of rat DRG (*Allodi et al., 2013*; *Santos et al., 2016a*; *Santos et al., 2016b*). In contrast, when we evaluated the neurite growth in each population separately, we saw that these factors had unique impacts. The neurite extension of proprioceptors was only promoted by NT-3. Contrarily, neither mech-anoreceptors nor nociceptors showed an improved neurite extension when cultured in presence of this factor. NT-3 has been described as a 'muscle factor' because of its trophic effect on proprioceptors and motoneurons (*Ernfors et al., 1995*; *Braun et al., 1996*; *Genç et al., 2004*; *Taylor et al., 2005*). Here, we confirmed that an overall trophic effect of NT-3 in sensory neurons is largely accounted for by neurite growth from proprioceptors. Nociceptors demonstrated a great increase in neurite length with the presence of NGF and, to a lesser extent, of BDNF. Mechanoreceptors showed a similar response pattern, although it was non-significant. Therefore, NGF could be considered a 'cutaneous factor', as it modulates specifically neurite extension of cutaneous neurons but not muscle neurons. In fact, NGF has been shown to act specifically on a subpopulation of small primary sensory neurons and on sympa-thetic neurons (*Levi-Montalcini, 1987*), and later studies confirmed that this factor increased sensory neurites, but not motor neurites (*Allodi et al., 2013*; *Santos et al., 2016b*). Thus, neurotrophic factors can be used to modulate differentially muscle and cutaneous sensory neurons.

Among the most DEGs, we found that Med12 was strongly upregulated in proprioceptors but downregulated in nociceptors and mechanoreceptors. Since the regeneration of proprioceptors is limited, we thought that axotomy might trigger the expression of regeneration inhibitory factors in this population that slows this process. MED12 is a subunit of Mediator, a multiprotein complex that regulates transcriptional activity (*Ding et al., 2008*). Besides its known involvement in genomic signaling, cytoplasmic MED12 was described to inhibit TGF-β receptor 2 through a physical interac-tion (*Huang et al., 2012*). The TGF-β pathway is a positive regulator of regeneration since it contrib-utes to the activation of the intrinsic growth capacity of neurons (*Walshe et al., 2011*; *Ye et al., 2022*). Knowing MED12's involvement in the TGF-β pathway, we hypothesized that the upregulation of this protein could hinder axon regeneration in proprioceptors through inhibition of TGF-β receptor 2. When silencing Med12, we found a significant increase in the length of neurites that was specific to proprioceptors, with no discernible impact on other sensory populations. Since nociceptors and mechanoreceptors showed a strong downregulation of *Med12*, we believe that silencing this gene has little effect on these cells. Altogether, our data demonstrates that MED12 is a novel regulator of neuron-specific regeneration and can be a promising factor to improve proprioceptive regeneration after nerve injury. The mechanism by which MED12 regulates this process is, however, unknown. We did not observe a change in expression of some typical TGF-β mediators. This does not exclude this pathway as the mechanism of action of MED12 since the low percentage of proprioceptive cells could be masking the effects. Future investigations should focus on elucidating the specific mechanisms by which MED12 modulates regeneration and neurite outgrowth. As a unique strategy to enhance proprioceptive neuron plasticity, in vivo analysis of the functional impact of its knockdown will be of significant interest.

Besides *Med12*, we found a large sample of genes significantly regulated after an injury, and many of them could explain the differential response seen in the regeneration of peripheral neuron subtypes. From these, we focused on the differences between muscle and cutaneous neurons since these could help us improve specific regeneration toward their target organs.

Muscle neurons strongly upregulated *Bcam* and *Myadm*, among other genes. *Bcam* encodes a member of the immunoglobulin superfamily that binds to laminin (*Udani et al., 1998*). Although this protein has been mainly studied in erythrocytes (*Wautier et al., 2007*; *Bartolucci et al., 2010*), laminin is the main substrate in the peripheral nerve and its differential expression might influence regeneration. MYADM has been described to be associated with lipid rafts (*Capkovic et al., 2008*; *Aranda et al., 2013*), which are membrane domains involved in growth factor signal transduction, axon guidance, and cellular adhesion (*Tsui-Pierchala et al., 2002*). In HeLa and PC3 cell cultures, MYADM is crucial for targeting RAC1 to these membrane rafts, facilitating cell migration (*Aranda et al., 2013*). Since RAC1 is one of the most important Rho GTPases favoring axon elongation, MYADM could be an interesting target for future regeneration studies. In contrast, one the genes more upregulated in cutaneous neurons compared to muscle neurons was *Gpc2*, which encodes for a cell surface proteoglycan. In N2a cells, the interaction of GPC2 with midkine (*Sorrelle et al., 2017*) was shown to promote cell adhesion and neurite outgrowth (*Kurosawa et al., 2001*). This suggests that GPC2 could be a target to enhance cutaneous specificity in mechanoreceptors and nociceptors.

Given the plethora of DEGs identified in a single neuron subtype, several could account for their distinct regeneration patterns. Among the top motoneuron-specific upregulated genes we found *Ngfr* (p75[NTR]), which was also found upregulated in previous motoneuron-specific RNA-sequencing (*Shadrach et al., 2021*). The functions of p75[NTR] are complex and diverse since its activation by neurotrophins can have opposite effects, including both survival and apoptosis (*Roux and Barker, 2002*; *Chao, 2003*; *Gutierrez and Davies, 2011*). NRP2 is a cell surface receptor for class 3 semaphorins, with high affinity to SEMA3C and SEMA3F (*Chen et al., 1997*). This receptor participates in axon guidance during development (*Giger et al., 2000*; *Gil and Del Río, 2019*), but its role in peripheral nerve injuries is largely unknown. In our study, we found a significant upregulation of *Nrp2* (and *Nrp1*) only in proprioceptors, the neuron population linked to less robust regenerative growth. Semaphorins are expressed in the nerve after an injury (*Ara et al., 2004*), and their repulsive effect through NRP2 could partially account for impaired growth in proprioceptive axons. In contrast, *Ndel1* was upregulated in all the studied neuron populations, except for proprioceptors. NDEL1 is a cytoskeleton integrator that participates in the formation of the vimentin-dynein complex during neurite extension (*Shim et al., 2008*). In vivo silencing of *Ndel1* after nerve injury resulted in reduced regeneration (*Toth et al., 2008*). Thus, the lack of upregulation of *Ndel1* in proprioceptors is in agreement with the limited regeneration observed in this population after a nerve injury.

Finally, the regenerative transcriptome from mechanoreceptors and nociceptors can also be useful to study candidates regulating allodynia and hyperalgesia after a nerve injury. We found a significant upregulation of *Atf2* in mechanoreceptors. Upregulation of this factor in small and medium DRG neurons was previously reported after injury, and its knockdown reduced tactile allodynia and thermal hyperalgesia after spinal nerve ligation (*Salinas-Abarca et al., 2018*). Moreover, we found a strong upregulation of *Il6ra* (interleukin-6 receptor alpha or gp80) in nociceptors. In neuropathic pain models, the proinflammatory cytokine interleukin-6 (IL-6) is involved in hyperalgesia (*Cunha et al., 1992*; *Murphy et al., 1999b*; *Arruda et al., 2000*; *Melemedjian et al., 2010*; *Quarta et al., 2011*; *Liu et al., 2019*). Furthermore, some studies support that IL-6 is involved in the pro-regenerative response of sensory neurons (*Hirota et al., 1996*; *Murphy et al., 1999a*; *Murphy et al., 2000*; *Cafferty et al., 2004*; *Cao et al., 2006*; *Zorina et al., 2010*). Thus, the high expression of *Il6ra* in nociceptors could be one of the contributing mechanisms to the greater regeneration of these neurons after nerve injury.

Altogether, we identified several DEGs not previously associated with the regenerative response. We also reported differences between neuron populations in genes that are known to be associated with a regenerative phenotype. Yet, it is worth noting that regenerative responses are dynamic, and we only analyzed the transcriptome 7 days after injury. Since we observed that regeneration rate varies between neurons, differences in gene expression could be influenced by the stage of regeneration. Analyzing different times after injury could help understanding some of the differences in the regenerative response of neuron populations. For instance, discriminating those set of genes that are specifically activated by a population after injury from the differential activation of the same set

of genes at different regeneration stages. Furthermore, we studied four neuron populations, some of which encompass more than one neuron subtype and not all of them are found in the same micro-environment. Motoneuron's somas are found in the spinal cord, further away from the lesion, and are influenced by a completely different environment than sensory neurons. Therefore, the direct comparison of motor and sensory neurons is limited and challenging. However, here we used a well-established model of injury that resembles the physiological conditions. Despite these caveats, we think that our findings are a key step to decipher the intrinsic growth capacity of different types of peripheral neurons.

## Conclusion

We characterized the regeneration of four key peripheral neuron subtypes from a histological and a genetic perspective. Our study aimed to identify specific mechanisms that could potentially be used in the future for modulating preferential regeneration of these neuron subtypes. Through our analysis, we were able to observe significant differences in the regeneration rate and gene expression after nerve injury that could be attributed to their distinct regenerative profiles. These findings provide valuable insights into the mechanisms driving the regeneration of different neuron subtypes and could serve as a basis for future studies focused on developing targeted approaches to promote specific types of regeneration.

# Materials and methods
## Animals

All experimental procedures were approved by the Universitat Autònoma de Barcelona Animal Experimentation Ethical Committee and followed the European Communities Council Directive 2010/63/EU and the Spanish National law (RD 53/2013). Mice were generated by breeding homozygous Ai9(R-CL-tdT) mice (JAX stock #007909) (*Madisen et al., 2010*) with different homozygous Cre-driver lines from The Jackson Laboratory (Bar Harbor, ME, USA): ChAT-IRES-Cre (choline acetyltransferase, JAX stock #006410) (*Rossi et al., 2011*), B6 PVcre (parvalbumin, JAX stock #017320), Npy2r-IRES-Cre (neuropeptide Y receptor Y2, JAX stock #029285) (*Barrozo et al., 2016*), and TRPV1-Cre (transient receptor potential vanilloid 1, JAX stock #017769) (*Cavanaugh et al., 2011*). We obtained four mice lines that expressed the red fluorescent protein TdTomato under the control of a specific neuronal promoter: ChAT-Cre/Ai9, PV-Cre/Ai9, Npy2r-Cre/Ai9, and TRPV1-Cre/Ai9, respectively. The same Cre-driver lines were bred to homozygous Ribotag mice (*Sanz et al., 2009*). We obtained mice that expressed HA-tagged ribosomes in specific cell populations: ChAT-Cre/Ribotag, PV-Cre/Ribotag, Npy2r-Cre/Ribotag, and TRPV1-Cre/Ribotag, respectively. Mice were housed in a controlled environment (12 hr light-dark cycle, 22 ± 2°C), in open cages with water and food ad libitum.

## Histological characterization of TRPV1-Cre/Ai9 mice

Three adult mice (8–12 weeks of age) were euthanized with intraperitoneal sodium pentobarbital (30 mg/kg) and perfused with 4% paraformaldehyde (PFA) in phosphate-buffered saline (PBS). Lumbar DRGs and footpads were harvested and stored in PBS containing 30% sucrose at 4°C for later processing. DRGs were serially cut in a cryostat (15 µm thick) and picked up on glass slides, whereas footpads were cut (50 µm thick) and stored free-floating. Samples were hydrated with PBS for 10 min and permeabilized with PBS with 0.3% Triton (PBST) for 10 min twice. Sections were blocked with 10% normal donkey serum in PBST for 1 hr at room temperature and then incubated with the

**Table 1.** Antibodies used for histological validation of TRPV1-Cre/Ai9.

| Sample | Thickness | Immunofluorescence |
|---|---|---|
| DRG | 15 µm | Parvalbumin (1:1000; Swant Cat# PV-28, RRID:AB_2315235)<br>Neurofilament H (1:1000; BioLegend Cat# 801701, RRID:AB_2715852)<br>CGRP (1:200; Millipore Cat# PC205L, RRID:AB_2068524)<br>Calbindin D-28K (1:200, Millipore Cat# AB1778, RRID:AB_2068336)<br>Isolectin B4 (10 µg/mL; Vector Laboratories Cat# L-1104, RRID:AB_2336498)<br>Anti-lectin I (1:500; Vector Laboratories Cat# AS-2104, RRID:AB_2314660) |
| Skin | 50 m | PGP 9.5 (1:500; Spring Bioscience Cat# E3340, RRID:AB_1661545) |

primary antibody in PBST overnight at 4°C (*Table 1*). Sections were washed with PBST three times and further incubated with a specific secondary antibody bound to Alexa 488 (1:200, Invitrogen) or Cy5 (1:200, Jackson ImmunoResearch) for 2 hr at room temperature (DRGs) or overnight at 4°C (skin). For IB4 immunostaining, an overnight incubation at 4°C with anti-lectin I (1:500; Vector, #AS2104) was done prior to the secondary antibody incubation. Finally, after three more washes in PBS, slides were mounted with Fluoromount-G mounting medium (Southern Biotech, 0100-01) and imaged with a confocal microscope (Leica SP5). In the DRG slices, we counted the number of sensory cells TdTomato-positive (TdTomato⁺) and how many of them co-labeled with the different markers using ImageJ software. We counted at least 929 cells per animal. For size distribution, we contoured 100 neurons of each type using ImageJ software. Skin was analyzed qualitatively, looking for the presence and distribution of fluorescence.

## Regeneration rate

Fifty-six adult mice (30 males and 26 females, 7–12 weeks of age) were used for establishing the regeneration rate of the four different neuron subpopulations. Mice were anesthetized with intraperitoneal ketamine (90 mg/kg) and xylazine (10 mg/kg) and the right sciatic nerve was exposed through a gluteal muscle-splitting incision. Nerves were crushed 3 mm distal to the sciatic notch with fine forceps (Dumont no. 5) applied for 30 s. The lesion site was labeled with an epineural suture stitch (10-0 nylon suture, Alcon) and the muscle and the skin were closed (6-0 nylon suture, Aragó). Animals were monitored periodically until the end of the experiments. After 7 or 9 days, mice were euthanized with intraperitoneal pentobarbital (30 mg/kg) and perfused with 4% PFA in PBS (n=7 for each day and Cre/Ai9 mice). The right sciatic nerve and its extension as tibial nerve were harvested and stored in PBS with 30% sucrose. We also collected some contralateral nerves as controls (n=6 ChAT-Cre/Ai9, n=3 PV-Cre/Ai9, n=3 Npy2r-Cre/Ai9, n=3 Trpv1-Cre/Ai9). A segment from 12 to 17 mm from the lesion site was serially cut in 10 µm longitudinal sections in a cryostat (Leica) and picked up in glass slides. The slides were mounted with Fluoromount-G mounting medium (SouthernBiotech, #0100-01) and visualized in an epifluorescence microscope (Nikon Eclipse Ni-E, Nikon, Tokyo, Japan) equipped with a digital camera (Nikon DS-RiE, Nikon, Tokyo, Japan) and Nikon NIS-Element BR software (version 5.11.03, Nikon, Tokyo, Japan). Using the software, we drew a line perpendicular to the nerve and counted the number of axons that crossed the line at 12 and 17 mm from the injury in one out of three sections.

## In vitro neurite extension

Explants and organotypic cultures were obtained from postnatal mice (p7-p8) as previously described (*Allodi et al., 2011*; *Bolívar et al., 2021*). Round coverslips were placed in 24-well plates and coated with poly-D-lysine (10 µg/mL) overnight at 37°C. Then, coverslips were washed three times with sterile distilled water and dried at room temperature. A 25 µL drop of collagen matrix composed by rat tail type I collagen solution (3.4 mg/mL; Corning, #354236) with 10% of minimum essential medium 10× (Gibco, #11430030) and 0.4% sodium bicarbonate (7.5%; Gibco, #25080-094) was placed on top of each coverslip. Plates were kept in the incubator at 37°C and 5% $CO_2$ for at least 2 hr to induce collagen gel formation.

PV-Cre/Ai9, Npy2r-Cre/Ai9, and TRPV1-Cre/Ai9 postnatal mice were sacrificed with intraperitoneal pentobarbital (30 mg/kg) and their lumbar DRGs were dissected and placed in cold Gey's solution (Sigma, #G9779) supplemented with 6 mg/mL glucose. Connective tissue was eliminated and DRGs were placed in the previously prepared collagen matrices. An additional 25 µL drop of collagen matrix was applied to cover the samples and the plates were left in the incubator for 45 min. Then, Neurobasal medium (Gibco, #21103049) supplemented with 1× B27 (Gibco, #17504044), 1× Glutamax (Gibco, #35050-038), 6 mg/mL glucose, and 1× penicillin/streptomycin (Sigma, #P0781) was added and plates were incubated at 37°C and 5% $CO_2$. After 2 days in culture, explants were fixed with warm 4% PFA in PBS for 30 min at room temperature. After washing with Tris-buffered saline (TBS), coverslips were detached from the plate and mounted onto glass slides using Fluoromount-G medium (Southern Biotech, #0100-01) and imaged with a confocal microscope (Leica SP5). All neurites in each DRG were semi-automatically measured in ImageJ, using the plugin SNT (Simple Neurite Tracer). Each individual DRG explant was treated as an experimental unit.

## Cre/Ribotag immunofluorescence

Control mice ChAT-Cre/Ribotag, PV-Cre/Ribotag, Npy2r-Cre/Ribotag, and TRPV1-Cre/Ribotag were perfused with 4% PFA in PBS. Lumbar DRGs were removed and stored in PBS containing 30% sucrose at 4°C for later processing. A lumbar segment of the spinal cord from ChAT-Cre/Ribotag mice was harvested and post-fixed in 4% PFA in PBS before storage. Samples were cut in a cryostat (15 µm thick) and processed for immunofluorescence as described above. Tissues were first incubated with rabbit anti-HA antibody (1:1000; Thermo Fisher Scientific Cat# 71-5500, RRID:AB_2533988) overnight at 4°C and then with anti-rabbit 488 (1:200, Molecular Probes Cat# A-21206, RRID:AB_2535792) for 2 hr at room temperature. Slides were imaged using an epifluorescence microscope (Olympus BX51, Olympus, Hamburg, Germany) equipped with a digital camera (Olympus DP50, Olympus, Hamburg, Germany).

## Ribotag assay

Fifty-six adult ChAT-Cre/Ribotag, PV-Cre/Ribotag, Npy2r-Cre/Ribotag, and TRPV1-Cre/Ribotag mice (total of 30 females and 26 males, 8–11 weeks of age) were anesthetized with intraperitoneal ketamine (90 mg/kg) and xylazine (10 mg/kg), and the right sciatic nerve was crushed as previously described. Then, the femoral nerve was exposed and crushed for 30 s above the bifurcation. The skin was closed with a 6-0 nylon suture (Aragó) and animals were monitored periodically until the end of the experiments. After 7 days, mice were euthanized with intraperitoneal pentobarbital (30 mg/kg), and L3, L4, and L5 DRGs (for PV-Cre/Ribotag, Npy2r-Cre/Ribotag, and TRPV1-Cre/Ribotag) or spinal cord (for ChAT-Cre/Ribotag) were dissected. Samples were placed on cold Gey's solution enriched with glucose (6 mg/mL) until they were used for the Ribotag assay. For isolation of sensory neurons RNA, L3-L5 DRGs from groups of three (Npy2r-Cre/Ribotag and TRPV1-Cre/Ribotag) or four mice (PV-Cre/Ribotag) were pooled and homogenized in 1 mL of homogenization buffer as described previously (*Sanz et al., 2009*). For motoneurons, the ipsilateral ventral horn of spinal cords from L3 to L5 from groups of three animals were pooled and homogenized in 1 mL of buffer. Female and male mice were used for the study, pooled in groups of the same sex. At least four pools of each condition and neuron type were used for the study (ChAT-Cre/Ribotag: 27 mice; PV-Cre/Ribotag: 36 mice; Npyr2-Cre/Ribotag: 24 mice; TRPV1-Cre/Ribotag: 24 mice) (*Table 2*). After centrifugation of the homogenate, 40 µL of the supernatant was stored as an input sample, whereas 4 µL of anti-HA antibody (Covance, #MMS-101R) was added to the remaining lysate and incubated for 4 hr at 4°C with rotation. Then, 200 µL of protein A/G magnetic beads (Thermo Fisher, #88803) were washed and added to the lysate for 5 hr (DRG samples) or overnight (spinal cords) at 4°C with rotation. Samples were washed in a high-salt buffer to remove non-specific binding from the IP and beads were pulled out using a magnet. RNA was isolated from the samples using the RNeasy Micro Kit (QIAGEN, #74004) and quantified with Quant-it RiboGreen RNA Assay Kit (Thermo Fisher, #R11490). The integrity of the RNA was assessed by using the RNA integrity number (RIN), an objective metric of total RNA quality ranging from 10 (highly intact RNA) to 1 (completely degraded RNA). RIN was obtained by the 2100 Bioanalyzer system with the RNA 6000 Nano or Pico chips (Agilent Technologies). All samples had RIN >6.

## qRT-PCR

1 µL of RNA was assayed using the TaqMan RNA-to-Ct 1-Step Kit (Thermo Fisher, #4392938). Specific transcripts were detected using Taqman assays: *Actb* (Mm02619580_g1), *Fabp7* (Mm00445225_m1), *Chat* (Mm01221882_m1), *Pvalb* (Mm00443100_m1), *Npy2r* (Mm01956783_s1), *Trpv1*

**Table 2.** Number of pools of each type used for the RNA-sequencing analysis.

| | Control | | | Crush | | |
|---|---|---|---|---|---|---|
| | *Female* | *Male* | *Total* | *Female* | *Male* | *Total* |
| ChAT-Cre/Ribotag | 3 | 2 | **5** | 2 | 2 | **4** |
| PV-Cre/Ribotag | 2 | 2 | **4** | 3 | 2 | **5** |
| Npy2r-Cre/Ribotag | 2 | 2 | **4** | 2 | 2 | **4** |
| TRPV1-Cre/Ribotag | 2 | 2 | **4** | 2 | 2 | **4** |

(Mm01246302_m1). Relative expression was obtained by normalizing to *Actb* RNA levels with the standard curve method.

## Library preparation and RNA-sequencing

The Genomics Core Facility (Universitat Pompeu Fabra) performed the RNA quality control and RNA-sequencing. Validity of the samples was assessed with 4200 TapeStation System (Agilent). 100 ng of total RNA from each sample was used for preparing the libraries with the NEBNext Ultra II Directional RNA Library Prep kit (New England Biolabs, #E7760), using the rRNA depletion module. Libraries were validated with TapeStation and all samples were pooled and sequenced using the NextSeq 500 System (Illumina) in runs of 2×75 cycles, yielding at least 30 million reads per sample. The raw data are available on the NCBI Sequence Read Archive (SRA) (accession PRJNA1101080).

FASTQ files were tested for quality using FastQC (v0.11.9). The reads were aligned to the coding DNA reference database (Ensembl Mouse database, Genome assembly: GRCm39, release 102) using Salmon (v1.8.0). The nature of the pair-end library was checked using Salmon, detecting an ISR type: inward, stranded, and read 1 coming from the reverse strand. The Salmon option 'validateMappings' was used. The quantified transcript reads were mapped to genes and imported to the R environment (R version 4.2.0) using the library 'tximport' (v1.24.0). Then, the 'DESeq2' library (v1.36.0) was used to perform the differential expression analysis. Volcano plots were plotted using the R library 'Enhanced-Volcano' (v1.14.0). Hierarchical clusters were performed using the R library 'pheatmap' (v 1.0.12) using the different groups analyzed in the study. In each case, the 80 more variable genes among samples were used, performing the clustering for both the rows (genes) and columns (samples). The enrichment analysis was performed using the R library 'STRINGdb' (v 2.8.4), using the genes quantified with an adjusted p-value lower than 0.05. Several ontologies and databases were used in this enrichment analysis: Biological Process (Gene Ontology), Molecular Function (Gene Ontology), Cellular Component (Gene Ontology), and KEGG Pathways.

## Immunohistochemistry against some factors specifically upregulated in subsets of neurons

Animals ChAT-Cre/Tomato, and Npy2r-Cre/Tomato, either control or 1 week after suffering a nerve crush, were perfused and lumbar DRGs and a lumbar segment of the SC were extracted. Samples were post-fixed in 4% PFA in PBS before storage. Samples were cut in a cryostat (15 μm thick) and processed for immunofluorescence as described above. Tissues were first incubated with rabbit anti-p75 antibody (1:100; Millipore Cat# AB1554, RRID:AB_90760) or rabbit ATF2 antibody (1:100, Thermo Fisher Scientific Cat# MA5-32022, RRID:AB_2809316) overnight at 4°C and then with anti-rabbit 488 (1:200, Molecular Probes Cat# A-21206, RRID:AB_2535792) for 2 hr at room temperature. Slides were imaged using an epifluorescence microscope (Olympus BX51, Olympus, Hamburg, Germany) equipped with a digital camera (Olympus DP50, Olympus, Hamburg, Germany).

## DRG explants and trophic factors

DRG explants were done as described above (in vitro neurite extension). For testing the effect of trophic factors in neurite extension, collagen matrices were enriched with either 10 ng/mL NGF (Peprotech, #450-01), or 10 ng/mL BDNF (Peprotech, #450-02), 10 ng/mL NT-3 (Peprotech, #450-03). Plates were kept in the incubator at 37°C and 5% $CO_2$ for at least 2 hr to induce collagen gel formation.

After 2 days in culture, explants were fixed with warm 4% PFA in PBS for 30 min at room temperature. After washing with TBS and TBS with 0.3% Triton (TBST), matrices were incubated with hot citrate buffer for 1 hr. Then, samples were incubated with 50%, 70%, and 100% methanol for 20 min each. Matrices were washed again and then incubated with rabbit anti-PGP9.5 (1:500, Spring Bioscience Cat# E3340, RRID:AB_1661545) in TBST and 1.5% normal donkey serum for 48 hr at 4°C. After three more washes, samples were incubated with anti-rabbit 488 (1:200, Molecular Probes Cat# A-21206, RRID:AB_2535792) in TBST and 1.5% normal donkey serum overnight at 4°C. Finally, matrices were washed, and the coverslips were detached from the plate. Coverslips were mounted onto glass slides using Fluoromount-G medium (Southern Biotech, #0100-01) and imaged with an epifluorescence microscope (Olympus BX51, Olympus, Hamburg, Germany) equipped with a digital camera (Olympus DP50, Olympus, Hamburg, Germany) and a confocal microscope (Leica SP5). The longest TdTomato$^+$

**Table 3.** Sequences of the siRNAs used in the cultures.

| siRNA | Sequence |
| --- | --- |
| | CCUAAGGUUAAGUCGCCCUCG |
| Scrambled | CGAGGGCGACUUAACCUUAGG |
| | AAGAACACCAUCUACUGUAAC |
| Med12_1 | GUUACAGUAGAUGGUGUUCUU |
| | AAGAACGUCAACUUCAAUCCU |
| Med12_2 | AGGAUUGAAGUUGACGUUCUU |
| | AAGCAGCUAAUGCAUGAGGCA |
| Med12_3 | UGCCUCAGCAUUAGCUGCUU |
| | AAGUGAAAGUGAGCGAGUAGA |
| Med12_4 | UCUACUCGCUCACUUUCACUU |

and PGP9.5$^+$ neurites of each DRG were measured using ImageJ software. Each individual DRG explant was treated as an experimental unit.

## DRG dissociated culture and siRNA

Adult male and female PV-Cre/Ai9 mice were sacrificed with intraperitoneal pentobarbital (30 mg/ kg) and all their DRGs were dissected and placed in cold Gey's solution (Sigma, #G9779) supplemented with 6 mg/mL glucose. DRGs were enzymatically dissociated in Ca and Mg free Hank's medium with 10% collagenase A (Sigma, #C2674), 10% trypsin (Sigma, #T-4674), and 10% DNAse (Roche, #11284932001) for 45 min at 37°C. Then, DRGs were mechanically digested by pipetting. Enzymes were inhibited with 10% hiFBS in DMEM (Sigma, #41966052) and centrifuged at 900 rpm for 7 min. Pellet was resuspended with 1 mL of Neurobasal-A (Gibco, #10088022) and filtered through a 70 μm cell mesh. The cell suspension was then carefully pipetted on top of 2 mL of 15% bovine serum albumin (Sigma, #A6003) in Neurobasal-A and centrifuged again. The pellet was resuspended in Neurobasal-A supplemented with 1× B27 (Gibco, #17504044), 6 mg/mL of glucose, 1× Glutamax (Gibco, #35050-038), and 1× penicillin/streptomycin (Sigma, #P0781). Then, either 8×10$^3$ cells (for immunostaining) or 30×10$^3$ cells (for qPCR) were plated in 24-well plates with medium containing siRNAs (Sigma) and HiPerFect transfection reagent (QIAGEN, #301704). The knockdown was achieved using four different siRNAs each at 50 nM (*Table 3*) whereas control wells contained scrambled siRNA at 200 nM. The transfection reagent and siRNAs were mixed at least 20 min before plating. Cells were cultured at 37°C and 5% CO$_2$ for 24 hr. Plates were previously coated with PDL (10 μg/mL; Sigma, #P6407) overnight and with laminin (1 μg/mL; Sigma, #L-2020) 2 hr at 37°C.

## Immunofluorescence of dissociated cultures

Cells were fixed with 4% PFA in PBS at room temperature for 20 min. After washing with PBS and permeabilizing with PBST, wells were incubated in 10% normal donkey serum diluted in PBST for 30 min. Then, wells were incubated overnight at 4°C with mouse anti-β-III-tubulin (1:500, BioLegend Cat# 801201, RRID:AB_2313773). After washing with PBST, cells were incubated with Alexa Fluor 488-conjugated anti-mouse antibody (1:200; Thermo Fisher Scientific Cat# A-21202, RRID:AB_141607) for 1 hr at room temperature and washed again with PBS. Coverslips were detached from the plates and mounted in glass slides with DAPI Fluoromount-G mounting medium (SouthernBiotech, #0100-20). We imaged neurons with an epifluorescence microscope (Olympus BX51, Olympus, Hamburg, Germany) equipped with a digital camera (Olympus DP50, Olympus, Hamburg, Germany). We imaged all TdTomato$^+$ neurons and at least 12 random fields in each coverslip for β-III-tubulin neurons. Neurite length was automatically analyzed using NeuroMath software. A total of four independent cultures were conducted and each one was treated as an experimental unit.

**Table 4.** Primers used for qPCR analysis.

| Gene | Direction | Sequence |
|---|---|---|
| | Forward | ctaaggccaaccgtgaaaag |
| Actin | Reverse | accagaggcatacagggaca |
| | Forward | agaaggttcaccaactgt |
| Med12 | Reverse | ctccttcttgaagatggaat |
| | Forward | cacagaggatcttgtcccccg |
| Smad7 | Reverse | ctggtctttcctcctgcgtt |
| | Forward | cacacccagctagggagactt |
| Crmp2 | Reverse | gtttaccccgtggtccttca |
| | Forward | tcctggagagtcctcagagc |
| Dab2 | Reverse | acctttgaacctggccaaca |
| | Forward | atcgctgcacccctttgagaa |
| Serpine | Reverse | atgcgggctgagatgacaaa |
| | Forward | aaatggaagcccagaaagatgc |
| Tgfbr2 | Reverse | tgcaggacttctggttgtcg |

## qPCR of dissociated cultures

Total RNA from cultures was isolated using the RNeasy Micro Kit (QIAGEN, #74004) and quantified using a NanoDrop. 200–300 ng of RNA were reverse-transcribed to cDNA using the High-capacity cDNA Reverse Transcription kit (Thermo Fisher, #4368814). For determining the expression of Med12, iTaq Universal SYBR Green supermix (Bio-Rad, #1725124) was used. The geometric mean of the expression levels of *Actb* was used to normalize the expression of Cp values. The primers used are listed in *Table 4*. Results from three independent cultures are reported and each culture was treated as an experimental unit.

## Data analysis

GraphPad Prism 8 (version 8.0.2) was used for statistical analysis. The normal distribution of the samples was tested with Shapiro-Wilk test. The statistical test used in each analysis is specified in the Results section. All data are expressed as group mean ± standard error of the mean (SEM). Differences were considered statistically significant if $p < 0.05$.

## Acknowledgements

The authors appreciate the technical help of Mònica Espejo, Jessica Jaramillo, and Neus Hernández. They also thank Honyi Ong and Sergi Verdés for the siRNA design, and the aid of José Manuel Crugeiras in the realization of some qPCR. This work was funded by the project SAF2017-84464-R, the grant FPU17/03657 from Ministerio de Ciencia, Innovación y Universidades of Spain, the grant PID2021-127626OB-I00 from Ministerio de Asuntos económicos y Transformación Digital of Spain and the Travel Grant from Boehringer Ingelheim Fonds. The author's research was also supported by funds from CIBERNED and TERCEL networks, co-funded by European Union (ERDF/ESF, 'Investing in your future').

# Additional information

## Funding

| Funder | Grant reference number | Author |
|---|---|---|
| Ministerio de Ciencia e Innovación | SAF2017-84464-R | Esther Udina |
| Ministerio de Ciencia e Innovación | FPU17/03657 | Sara Bolívar |
| Ministerio de Asuntos Económicos y Transformación Digital, Gobierno de España | PID2021-127626OB-I00 | Esther Udina |

The funders had no role in study design, data collection and interpretation, or the decision to submit the work for publication.

## Author contributions

Sara Bolívar, Conceptualization, Data curation, Software, Formal analysis, Validation, Investigation, Visualization, Methodology, Writing - original draft, Writing – review and editing; Elisenda Sanz, Douglas W Zochodne, Supervision, Investigation, Methodology, Writing – review and editing; David Ovelleiro, Data curation, Software, Formal analysis, Methodology, Writing – review and editing; Esther Udina, Conceptualization, Resources, Data curation, Software, Formal analysis, Supervision, Funding acquisition, Validation, Investigation, Visualization, Methodology, Project administration, Writing – review and editing

## Author ORCIDs

Sara Bolívar ⓘ http://orcid.org/0000-0003-2966-6845
Elisenda Sanz ⓘ http://orcid.org/0000-0002-7932-8556
Esther Udina ⓘ http://orcid.org/0000-0003-1954-8562

## Ethics

All experimental procedures were approved by the Universitat Autònoma de Barcelona Animal Experimentation Ethical Committee and followed the European Communities Council Directive 2010/63/EU and the Spanish National law (RD 53/2013).

Joint Public Review: https://doi.org/10.7554/eLife.91316.3.sa1
Author response https://doi.org/10.7554/eLife.91316.3.sa2

---

# Additional files

## Supplementary files

• Supplementary file 1. Expression of neurotrophin and GDNF receptors in immunoprecipitates compared to inputs.

• Supplementary file 2. Differentially expressed gene (DEG) in each neuron subpopulation.

• Supplementary file 3. Pathway enrichment of each neuron subpopulation in response to injury.

• MDAR checklist

## Data availability

Sequencing data have been deposited in SRA under accession code PRJNA1101080 https://www.ncbi.nlm.nih.gov/bioproject/?term=PRJNA1101080.

The following dataset was generated:

| Author(s) | Year | Dataset title | Dataset URL | Database and Identifier |
|---|---|---|---|---|
| Bolivar S, Udina E | 2024 | RNA-sequencing of different peripheral neurons after crush injury in mice | https://www.ncbi.nlm.nih.gov/bioproject/?term=PRJNA1101080 | NCBI BioProject, PRJNA1101080 |

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
