## [Editor Report · eLife assessment]

The **valuable** findings in this study show that subpopulations of peripheral sensory neurons display different capacities for regeneration after a similar injury. Nociceptor neurons have greater regeneration over mechanoreceptor, proprioceptors and motor neurons. This differential responsiveness of neuronal subtypes was traced to activation of different transcriptional programs, which were carefully analyzed and quantitated, resulting in **solid** evidence for the conclusions.

---

## [Referee Report · Joint Public Review]

Bolivar et al. set out to explore whether four distinct neuronal subtypes within the peripheral nervous system exhibit varying potentials for axon regeneration following nerve injury. To investigate this question, they harnessed the power of four distinct reporter mouse models featuring fluorescent labeling of these neuronal subtypes. Their findings reveal that axons of nociceptor neurons exhibit faster regeneration than those of motor neurons, with mechanoreceptors, and proprioceptors displaying the slowest regeneration rate.

To delve into the molecular mechanisms underlying this divergence in regeneration potential, the authors employed the Ribotag technique in mice. This approach enabled them to dissect the differential translatomes of these four neuronal populations after nerve injury, comparing them to uninjured neurons. Their comprehensive expression profiling data uncovers a remarkably heterogeneous response among these neuron subtypes to axon injury.

To focus on one identified target with a mechanistic experiment as a proof of concept, their analysis highlights a striking upregulation of MED12 in proprioceptors, leading to the hypothesis that this molecule may play an inhibitory role, contributing to the comparatively slower regeneration of proprioceptor axons when compared to other neuronal subtypes. This hypothesis gains support from their in vitro model, where siRNA-mediated downregulation of MED12 results in a significant increase in neurite outgrowth in proprioceptive neurons after plating in cell culture dishes.

Overall, this is an interesting study, and in conjunction with similar work from others will be highly valuable for neurobiologists studying how to modulate the regeneration of axons from distinct neuronal subtypes. The quality of data presentation appears to be very good in general, and the manuscript is appropriately written.

Comments on revised version:

Because there are multiple explanations for the differential regeneration responses, the authors have provided further discussion about how regeneration may be regulated in vitro and in vivo. The detection of a gene, Med12, which is unregulated in proprioceptive neurons, but not nociceptive and mechanoceptors, gives support to the existence of specific programs of responses in the peripheral nervous system after injury. Further investigation is needed to define this responsiveness in detail.

Another response is the role of neurotrophins and their receptors. The authors have considered outcomes as a result of different Trk receptor signaling and also the effect of TGFbeta and IL6 as cytokine modulators. Add to this list is the possibility that axon guidance molecules and downstream substates may also play a role.

The original title was considered to be too broad and did not explain all the mechanistic aspects of this study. Therefore a revised title "Neuron-specific RNA-sequencing reveals different response in peripheral neurons after nerve injury" was used. It is appropriately suitable for the results reported in this manuscript.

---

## [Author Response]

The following is the authors’ response to the original reviews.

We thank the reviewers and editors for their comments, as well as for the time dedicated to make useful suggestions that have contributed to improve the manuscript. We have responded to the concerns raised by the reviewers, and after that, we have also responded to the different points highlighted in the Recommendations for the authors:

**Reviewer #1**
While in vivo injury was used to assess regeneration from subsets of PNS neurons, different in vitro neurite growth or explant assays were used for further assessments. However, the authors did not assess whether the differential "regenerative" responses in vivo could be recapitulated in vitro. Such results will be important in interpreting the results.

We included a supplementary figure evaluating the neurite extension in vitro and updated the text accordingly.

Intriguingly, even in individual groups of PNS neurons, not all neurons regenerate to the same extent. It is known that the distance between the cell body and the lesion site affects neuronal injury responses. It would be interesting to test this in the observed regeneration.

Although it is true that the distance can affect the outcome, here we used a physiological model where all neurons are lesioned at the same point in the nerve. Not only distance is different for motoneurons, but also the microenvironment surrounding their somas and therefore the direct comparison of these neurons with sensory neurons is limited. We extended the discussion on this matter in the new manuscript.

Fig 1: The authors quantified the number of regenerating axons at two different time points. However, the total numbers of neurons/axons in each subset are different. The authors should use these numbers to normalize their regenerative axons.

Figure 1D shows the normalization of data from figure 1C (normalized against the number of control axons in each neuron type). This has been clarified in the text.

Fig 2-5: In explaining differential regeneration of individual groups of neurons, there are at least two possibilities: (1). Each group of neurons has different injury/regenerative responses; (2). The same set of injury/regenerative responses are differentially activated. Some data in this manuscript suggested the latter possibility. But some other data point in the opposite direction. It would be informative for the authors to analyze/discuss this further.

From our point of view, these two options can be considered differential response to injury and could be potentially used for the modulation of regeneration. However, if the second possibility is correct, the regenerative program could be more influenced by the time chosen to study the response. Given the importance of this, we added some discussion about this topic.

Fig 6: Is it possible to assess the regenerative effects of knockdown Med12 after in vivo injury?

It is possible, but it is out of the scope of this work. Here, we aimed to describe the regenerative response and validate our data by testing a potential target for specific regeneration. Future studies will focus on the modulation of this specific regeneration both in vitro and in vivo.

**Reviewer #2**
It seems that the most intriguing outcome of this paper revolves around the role of Med12 in nerve regeneration. The authors should prioritize this finding. Drawing a conclusion regarding Med12's role in proprioceptor regeneration based solely on this in vitro model may be insufficient. This noteworthy result requires further investigation using more animal models of nerve regeneration.

The main goal of this work was to compare the regenerative responses of different neuron subpopulations. We modulated Med12 to validate our data and the potential of our findings. Unfortunately, investigating in depth the role of Med12 in regeneration is out of the scope of this paper. For this reason, we did not prioritise this finding here. As this finding was striking, we strongly agree that the next step should be studying how it modulates regeneration.

One critique revolves around the authors' examination of only a single time point within the dynamic and continuously evolving process of regeneration/reinnervation. Given that this process is characterized by dynamic changes, some of which may not be directly associated with active axon growth during regeneration, and encompasses a wide range of molecular alterations throughout reinnervation, concentrating solely on a single time point could result in the omission of critical molecular events.

We agree that this is probably the main limitation of this study, as we discussed in the text. We chose 7 days postinjury as a standard time point widely described in literature and to have a correlate with our histological data. Although the main aim was to compare populations, analyzing an additional time point after injury could add valuable information.

**Reviewer #3**

No concerns were expressed by that reviewer.

**Recommendations for the authors:**
The authors should assess whether the differential "regenerative" responses in vivo could be recapitulated in vitro.

We included a supplementary figure evaluating the neurite extension in vitro and updated the text accordingly.

Optional:It will be interesting to test if the distances between the cell body and the lesion site contribute to the observed differences in individual subsets of PNS neurons.

Figure 1D shows the normalization of data from figure 1C (normalized against the number of control axons in each neuron type). This has been clarified in the text.

Fig 2-5: In explaining differential regeneration of individual groups of neurons, there are at least two possibilities: (1). Each group of neurons has different injury/regenerative responses; (2). The same set of injury/regenerative responses are differentially activated. Some data in this manuscript suggested the latter possibility. But some other data point in the opposite direction. At least the authors should discuss these.

From our point of view, these two options can be considered differential response to injury and could be potentially used for the modulation of regeneration. However, if the second possibility is correct, the regenerative program could be more influenced by the time chosen to study the response. Given the importance of this, we added some discussion about this topic.

While the paper is technically well-executed, the conclusions and some of the findings appear to be incomplete and challenging to draw meaningful conclusions from. This manuscript presents some interesting findings, but the title is quite broad and may suggest that the authors have unveiled fundamental mechanisms explaining the varying regenerative abilities of peripheral axons. However, the results do not substantiate such a conclusion. Further comments and suggestions follow.

We eliminated the word “regenerative (response)” from the title, as it could lead to think that all changes seen in these neurons are related only to regeneration. We think that “Neuron-specific RNA-sequencing reveals different responses in peripheral neurons after nerve injury” highlights the differences between neurons that we found without misleading towards thinking that we described regenerative mechanisms in all neurons.

What's notably absent here is the validation of certain genes found with the ribosomes, especially those highlighted in the subsequent figures. The question arises as to whether the changes depicted in the figures align with changes in the DRGs in vivo. Is there concordance between the presence of these genes and their transcriptional changes? It would greatly enhance the study's value if the authors could show evidence of upregulation or downregulation of certain genes over time in tissue sections, utilizing techniques such as in situ hybridization or immunocytochemistry.

We selected some factors that were specifically upregulated in subsets of neurons to corroborated by immunohistochemistry these findings. Changes in the immunofluorescence of P75 in motoneurons and ATF2 in cutaneous mechanoreceptors, were evaluated in controls and animals that received a nerve crush one week before. Supplementary figures with the images have been added.

The authors discovered intriguing distinctions, such as the presence of specific signaling pathways unique to neurons projecting to muscle as opposed to those projecting to the skin. Among these pathways were those associated with receptor tyrosine kinases like VEGF, erbB, and neurotrophin signaling among others. The question now arises: do these pathways play a role in natural peripheral regeneration processes? To answer this, it is imperative to conduct in vivo studies. However, the authors employed an in vitro DRG neurite outgrowth assay to demonstrate that various types of neurons exhibit different responses to the presence of different neurotrophins. This does not reflect what actually happens in vivo. While neurotrophins indeed play a role in neuron survival and axon extension during development, their role in postnatal periods changes over time, and it remains unclear whether they play any role in the natural regenerative processes of the peripheral nerve. Therefore, this experiment may not be directly relevant in this case, especially during the early axon extension period of the regenerating axons. if the authors aim to establish a causal link with neurotrophin signaling, it becomes crucial to conduct in vivo experiments by manipulating the expression of key molecules like the receptors.

It has been widely described that different types of peripheral neurons have a differential expression of Trk receptors, even in the adult, and that these respond differentially to neurotrophins. In our study, we do not stablish a causal relationship between the expression of Trk and neurite extension, but instead we show (as many others) that distinct neurons respond differentially to these neurotrophins. The fact that in vivo studies fail to show a clear effect does not necessarily mean that neurotrophins are not specific. It might mean that their effect is not strong enough to be a useful guide in the complex microenvironment found after an injury. For instance, NGF acts on TrkA (present in some neurons), but in vivo it has been shown to accelerate the clearance of myelin debris in Schwann cells (Li et al., 2020), which could facilitate regeneration of all type of axons, masking any potential specific effect on the subtypes of neurons expressing TrkA. In contrast, in an in vitro setting on neuronal cultures, the specific neuronal effect can be more evident.

Additionally, it's worth noting that another paper utilizing the same methodology and experimental setup (PMID: 29756027, "Translatome Regulation in Neuronal Injury and Axon Regrowth" by Rozenbaum et al.) exists. Are there any significant differences or shared findings with that study?

This study shows the transcriptomic response after an injury 4, 12 and 24 hours after an injury in a very similar experimental setup. They focus on comparing the neuronal vs the glial response to the injury, using a Ribotag line that tags ribosomes from all neurons in the DRG rather than specific neuron subtypes. As the time postinjury (24h vs 7 days) and the cell types studied are different, we could not directly compare our results. We did see an upregulation in both datasets of previously described growth-associated genes (Jun, Atf3, Sox11, Sprr1a, Gal…). We included the article in the references for its relevance in the topic.

It would be helpful for readers to illustrate the finding of the fastest axon regeneration of nociceptors by showing fluorescence micrographs of the nerve samples in addition to the graphs shown in Fig. 1 C/D.

In figure 1B, we show fluorescence micrographs of the nerves 7 days postinjury. As explained in the results, we counted the number of axons at 2 distances from the injury, we did not analyse the fastest axon. This is due to technical reasons: 7 days after the injury the fastest axon has surpassed our evaluation point, which was the further distance that we could assess in our experimental setting in a consistent manner. If the reviewer thinks that we need to include more images from our evaluations (from 9 dpi for example), we could prepare a new figure.

The labeling in Fig. 2B is confusing. Is the CHAT immunoreactivity shown in the last panel illustrated by green or red signals? Is the red signal counterstaining with beta-tubulin?

The labelling was changed in the figure to increase clarity.

The references to the supplementary data throughout the manuscript are confusing. For example, where can the "Supp data 2" be found? (mention on p. 14 in the merged pdf file). Are they referring to the Excel spreadsheets?

We divided the supplementary material in supplementary figures/table (found in the pdf) and supplementary data. Supplementary data refers to excel spreadsheets found outside the pdf file. We hope this will be clearer after the final formatting of the article.

What does the following statement on p. 14 mean?: "The caveat in these analyses was that molecular classification by these approaches may be arbitrary, and not reflective of protein repurposing." This reviewer notes that these databases consider the fact that components participate in different pathways.

Indeed, we aimed to explain that many proteins participate in different pathways, and this is a limitation of the enrichment analysis. We modified the sentence in the text.

First paragraph on p. 15: The PPAR and AMPK pathways have much broader roles, and are not only "related to fatty acid metabolism". This factual inaccuracy should be corrected in the manuscript.

The sentence has been corrected.

The authors should consider showing increased TGF-beta signaling in their neurons after downregulation of Med12 given the previous implication of TGF-beta signaling in axon regeneration.

We tried to demonstrate the effect of our knockdown in TGF-beta pathway by analyzing the expression of typical targets from this pathway by qPCR in our cultures. However, we could not detect any difference. We think that this can have two explanations: (1) as only a few cells upregulate Med12 whereas many cells downregulate it, the effect is masked (presumably only proprioceptors will have a significant difference in this pathway and, thus, it would be very difficult to see the effect), or (2) Med12 is not exerting its effect through this pathway. We added a supplementary figure with these data and discussed it in the manuscript.

It would be helpful to eliminate typos and improve syntax/grammar/style.

We revised the text to improve style.